# GAMMA SAMPLING: FINE-GRAINED CONTROLLING LANGUAGE MODELS WITHOUT TRAINING

## ABSTRACT

The dominant approaches for controlling language models achieve prominence in controlling high-level attributes (e.g., topic and sentiment). However, these methods often require condition-specific data or are computationally expensive. We propose a new simple guided decoding method, GAMMA SAMPLING, which does not require any training data to achieve fine-grained controllable text generation while maintaining a fast generation speed. GAMMA SAMPLING introduces attribute-related information (provided by humans or language models themselves) into the sampling process to guide language models to generate texts with desired attributes. Since no training is involved, GAMMA SAMPLING can be easily applied to any language model for controllable text generation. Through experiments, we show that GAMMA SAMPLING-steered GPT2 generally outperforms all the representative baselines for controllable generation in terms of diversity, attribute relevance, and overall quality of generated samples.

## 1 INTRODUCTION

Benefiting from large-scale text data crawled from the web, the state-of-the-art large language models (LMs) achieve great success in language generation. However, although existing models can generate high-quality texts, we have little control over the attributes (e.g., topic and sentiment) of generated outputs. This limitation makes it difficult to apply unconditional LMs to scenarios that require good control over the generated text. How to steer unconditional LMs, i.e., controllable text generation, becomes a topic of real-world significance.

Despite great advances in controllable text generation (Weng 2021), it remains an open question what the ideal method for controlling the attributes of the generated language is (Mireshghallah et al. 2022, Yang & Klein 2021, Meng et al. 2022). No matter whether training a conditional LM from scratch (Keskar et al. 2019) or fine-tuning an LM (Ziegler et al. 2019, Xu et al. 2021), the need for condition-specific data makes these approaches not easily applicable to unconditional LMs. On the other hand, although some methods (Shin et al. 2020, Zou et al. 2021, Ghazvininejad et al. 2017, Pascual et al. 2020, Lu et al. 2021) are data-free, they are often very limited in steerability or computationally intensive.

In this paper, we propose GAMMA SAMPLING for fine-grained controlling LMs, which does not require any training data and is computationally efficient. This method is inspired by gamma correction (Applebaum 1952), a nonlinear operation used to encode and decode luminance in video or still image systems. The basic assumption of GAMMA SAMPLING is that some attributes of the generated text are closely related to the occurrences of certain tokens. Therefore, we can **increase or decrease the probability of these attribute-related tokens to control the attributes of generated text**. Our key contributions are as follows.

- GAMMA SAMPLING, as a data-free approach, requires no training on LM or additional discriminators. It can be readily used to achieve controllable text generation for any LM by selecting attribute-related tokens manually or automatically.

- GAMMA SAMPLING supports combinations of multiple controllable attributes, and its control strength is fine-grained. Users can determine how strong the attribute relevance should be, and the control can be turned off at any time.

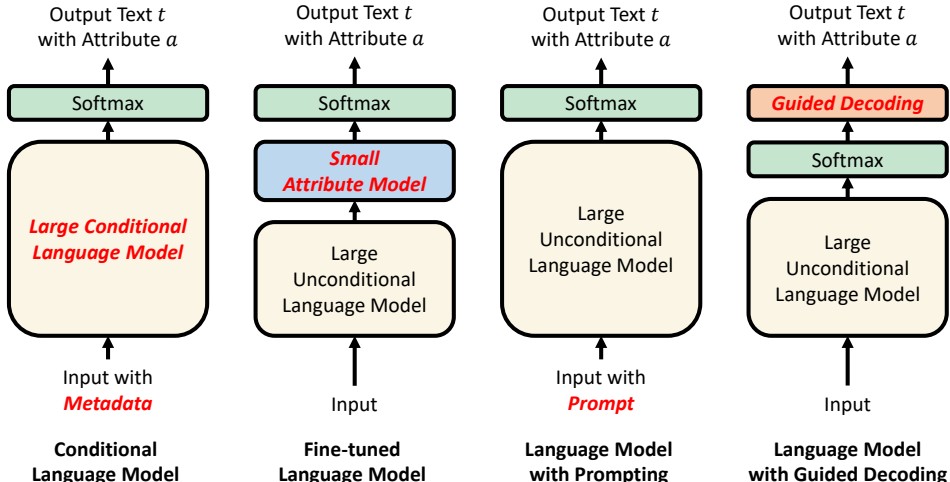

Figure 1: Common methods of controllable generation. Modules that introduce attribute-related information are marked in red.

- GAMMA SAMPLING is computationally efficient, as it requires only a slight modification at decoding time. Unlike other guided decoding methods (Pascual et al. 2021), the time cost of GAMMA SAMPLING is insensitive to the number of controlled tokens. Compared to PPLM (Dathathri et al. 2020), the generation speed of GAMMA SAMPLING is at least $100\times$ faster.
- We compared GAMMA SAMPLING-steered GPT2 in all sizes (i.e., Small, Medium, Large, and XL) with several common methods for controllable generation through both automatic and human evaluations. The results show that our method generally outperformed all the baselines in terms of diversity, attribute relevance, and overall quality.

## 2 BACKGROUND

The typical controllable generation is about modelling a probabilistic model $p(x|a)$, that is, generating text $x$ based on an attribute $a$. In contrast, for unconditional LMs, only $p(x)$ can be obtained directly. However, by using certain methods for controlling unconditional LMs, it is still possible to make the generated text $x$ have an attribute $a$. There are several common approaches for controllable generation, each with its pros and cons.

**Conditional Language Model** Conditions are introduced to models during training phases, which can be obtained from the metadata of the training data. However, as the entire LM needs to be trained from scratch, it requires a large amount of condition-specific data as well as considerable training costs. Furthermore, conditional LMs such as CTRL (Keskar et al. 2019) fall short in controlling what not to generate, e.g., detoxification and anti-degeneration (Gehman et al. 2020, Ma et al. 2020).

**Fine-tuned Language Model** Fine-tuned LMs (Ziegler et al. 2019, Xu et al. 2021) usually strike a good balance between training cost and generation quality. These models are based on existing large models with all the weights in them fine-tuned, limiting the fine-tuning to the top or additional layers only, or introducing discriminators (Krause et al. 2021, Liu et al. 2021a). However, fine-tuned LMs still require condition-specific data. Furthermore, models such as PPLM (Dathathri et al. 2020), which combine multiple small attribute models with a large LM, could cause computational efficiency to become unacceptable by requiring multiple passes at every decoding step.

**Prompting** As large LMs, e.g., GPT-2 (Radford et al. 2019) and GPT-3 (Brown et al. 2020), are trained on huge amounts of data, they are very powerful on many NLP tasks. By selecting appropriate prompts (Shin et al. 2020, Zou et al. 2021), unconditional LMs can be used to solve a wide range of downstream tasks. Although prompt engineering has become a recent research hotspot (Liu et al. 2021b), minor differences in prompt usually have a big impact on the performance of downstream tasks (Kojima et al. 2022).

**Guided Decoding** Although decoding does not affect any trainable parameters of LMs, it is a critical part of text generation. Guided decoding (Ghazvininejad et al. 2017, Ghosh et al. 2017, Pascual et al. 2020, Lu et al. 2021, Liu et al. 2021a, Pascual et al. 2021) introduces attribute-related

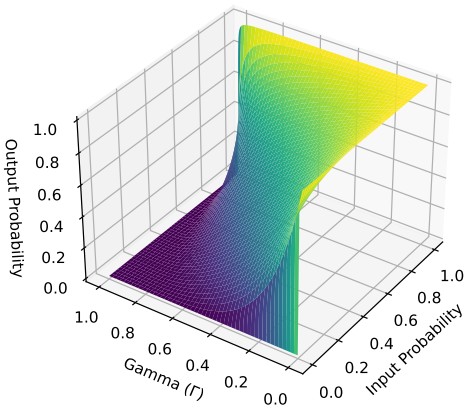
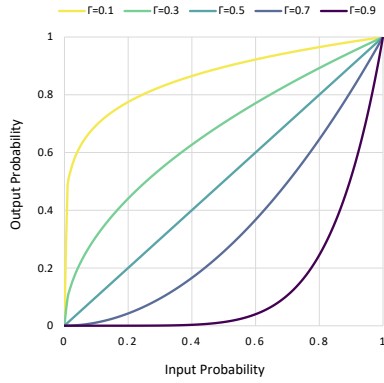

(a) 3D surface plot of GAMMA SAMPLING          (b) GAMMA SAMPLING with different $\Gamma$

Figure 2: The probabilities of selected tokens are scaled up when $\Gamma < 0.5$, scaled down when $\Gamma > 0.5$, and unchanged when $\Gamma = 0.5$.

information at the decoding time, thus allowing users' preferences on attributes to be injected into the score function to steer the sample generation by modifying the candidate ranking score based on model output logits (before the softmax layer). The main disadvantage of guided decoding is that it has the potential to go off-distribution when the guides are too strong. In addition, the time cost of guided decoding typically increases linearly with the number of controlled tokens. This results in unacceptable inference times when there are a large number of controlled tokens. Sec. 3.2.1 demonstrates that, although GAMMA SAMPLING is a guided decoding method, it does not have the above disadvantages (or at least can be avoided).

As shown in Fig. 1, regardless of what approaches are taken, LMs need to be provided with attribute-related information to achieve controllable generation. The key intuition behind the first two approaches is that, after considerable training, LMs can learn attribute-related information from large amounts of data and then use it to achieve controllable generation. On the other hand, the last two assume that large LMs already learned enough information from data and only need to introduce attribute-related information into the input or decoding to be applied to various downstream tasks.

## 3 METHODOLOGY

We describe GAMMA SAMPLING, a guided decoding method motivated by gamma correction, for controlling unconditional LMs. Sec. 3.1 briefly introduces gamma correction. Sec. 3.2 explains the details of GAMMA SAMPLING. Finally, Sec. 3.3 presents several implemented controllable attributes based on GAMMA SAMPLING that are defined by certain attribute-related tokens.

### 3.1 GAMMA CORRECTION

Although gamma correction (Applebaum 1952) was originally developed to compensate for the input-output characteristic of Cathode Ray Tube (CRT) displays, it is now also used to remodel the saturation of images. Gamma correction is responsible for performing a power function on all the pixels of the input image, as the human perception of brightness is nonlinear (Neri 2009). In the simplest cases, it is defined by the following power-law expression:

$$V_{out} = AV_{in}^{\gamma}, \tag{1}$$

where the luminance value $V_{in} \in [0, 1]$ is raised to the power $\gamma \in [0, +\infty)$ and multiplied by the constant $A$ (in the common case, $A = 1$) to get the output value $V_{out} \in [0, 1]$.

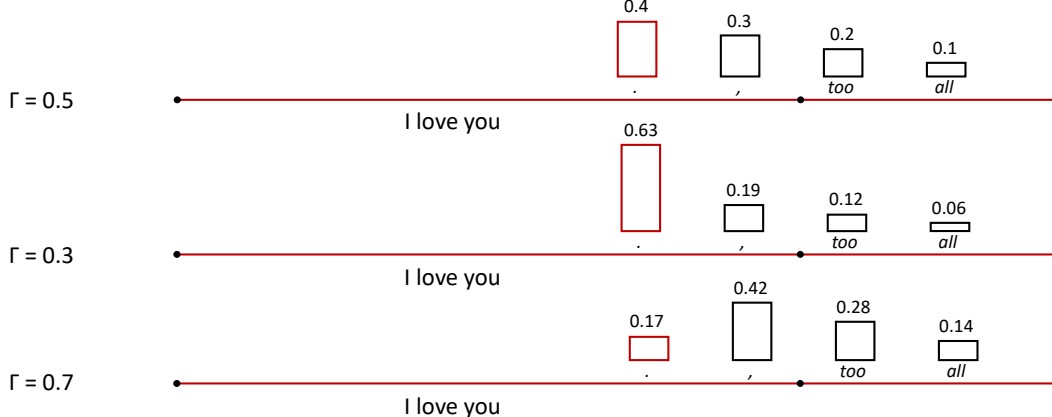

Figure 3: This example demonstrates controllable sentence length with the input `I love you`, where the full stop is the attribute-related token. The smaller the probability of a full stop being sampled ($\Gamma > 0.5$), the smaller the number of full stops in the generated text and the longer the average sentence length, and vice versa. See Sec. 4.2 for details.

## 3.2 GAMMA SAMPLING

### 3.2.1 BASIC PRINCIPLES

Similar to how humans perceive light and colour in a non-linear way (Neri 2009), the frequency of any word in common natural language corpora is inversely proportional to its ranking in the frequency table (Powers 1998). Therefore, it is reasonable to perform non-linear processes on the probabilities of attribute-related tokens. Temperature sampling (Dabre & Fujita 2021), as an example that is widely used in text generation tasks, is also done via a non-linear operation to scale the probability distribution.

We expect that just as gamma correction can fine-grained tune the luminance of images, a similar approach can be used to control the attributes of text. For probability distributions, the key non-linear operation we perform can be formalised as follows:

$$p_{\mathcal{A}_{out}} = p_{\mathcal{A}_{in}}^{tan(\frac{\pi\Gamma}{2})}, \tag{2}$$

where $p_{\mathcal{A}_{in}}$ is the sum of the input probabilities of all tokens in the attribute-related token set $\mathcal{A}$, $p_{\mathcal{A}_{out}}$ is the output one, and $\Gamma \in [0, 1]$ is the control strength. The major differences between GAMMA SAMPLING and gamma correction are as follows: 1) the non-linear operation of GAMMA SAMPLING is limited to attribute-related tokens used to introduce attribute-related information and is therefore not a global modification, and 2) we introduced the tan function to modify the range of the control strength. As shown in Fig. 2, the intervals in which the power-law expression for GAMMA SAMPLING is concave or convex are of equal range, which is much more user-friendly to tune.

It is important to note that using GAMMA SAMPLING on its own can be risky. When we set $\Gamma$ to a very small value, even if some attribute-related tokens have negligible probabilities (i.e., it is basically impossible for them to appear in the current context), GAMMA SAMPLING will greatly increase their probabilities regardless, which likely leads to off-distribution. Therefore, **it is necessary to use either top-$k$ sampling** (Fan et al. 2018) **or nucleus sampling** (Holtzman et al. 2020) **to truncate those unreliable tokens before GAMMA SAMPLING.**

GAMMA SAMPLING differs from other guided decoding methods in that: 1) it operates on probability distributions (after the softmax layer) rather than logits (Liu et al. 2021a), which allows top-$k$ sampling or nucleus sampling to be applied before it to avoid over-control strengths resulting in off-distribution; 2) there is no need to tailor the score function (Ghazvininejad et al. 2017), train additional discriminators (Krause et al. 2021), or use specific external models (Pascual et al. 2021), and only attribute-related tokens and the control strength $\Gamma$ need to be specified for GAMMA SAMPLING; 3) since the probability distribution is simply modified, it is not only computationally efficient, but the time cost is independent of the number of controlled tokens.

### 3.2.2 CONTROLLING SINGLE ATTRIBUTE

After scaling up or down the probabilities of attribute-related tokens, some post-processing needs to be done: 1) the sum of the probabilities of the attribute-related tokens can be scaled up to 1 or down to 0 to ensure a high degree of steerability, 2) the ratio of probabilities between tokens of the same type (i.e., attribute-related tokens or non-attribute-related tokens) remains unchanged, and 3) the output probability distribution is still valid. The processes for controlling a single attribute using GAMMA SAMPLING can be formalised as follows:

$$
\begin{aligned}
p_{\mathcal{A}_{out}} &= p_{\mathcal{A}_{in}}^{tan(\frac{\pi\Gamma}{2})}, \\
p_{a_{out}} &= p_{a_{in}} \cdot \frac{p_{\mathcal{A}_{out}}}{p_{\mathcal{A}_{in}}}, \quad \forall a \in \mathcal{A}, \\
p_{n_{out}} &= p^{n_{in}} \cdot (1 + \frac{p_{\mathcal{A}_{in}} - p_{\mathcal{A}_{out}}}{p_{\backslash \mathcal{A}_{in}}}), \quad \forall n \notin \mathcal{A},
\end{aligned}
\tag{3}
$$

where $p_{a_{in}}$ is the input probability of an attribute-related token $a$, $p_{\mathcal{A}_{out}}$ is the output one, and the same goes for every non-attribute-related token $n \in \backslash\mathcal{A}$. Eq. 3 first modifies the sum of the input probabilities of all attribute-related tokens $p_{\mathcal{A}_{in}}$, then rescales each attribute-related token $a$ by the ratio of $p_{\mathcal{A}_{out}}$ to $p_{\mathcal{A}_{in}}$, and finally rescales the probability of each non-attribute-related token $n$ by the ratio of the difference between $p_{\mathcal{A}_{in}}$ and $p_{\mathcal{A}_{out}}$ to the sum of the input probabilities of all non-attribute-related tokens $p_{\backslash\mathcal{A}_{in}}$. Fig. 3 visualises a simple example of language generation with GAMMA SAMPLING.

### 3.2.3 CONTROLLING MULTIPLE ATTRIBUTES

Assuming one needs to control $T$ attributes $\mathcal{A}^T = \{\mathcal{A}^1, \mathcal{A}^2, ..., \mathcal{A}^T\}$ in a sequential manner, simply performing Eq. 3 in order could lead to the earlier modifications being overwritten by the later ones. To solve this problem, we made further modifications based on the control of a single attribute. When controlling the $t$-th attribute $\mathcal{A}^t$, there are the following processes:

$$
\begin{aligned}
\mathcal{F}^t &= \mathcal{A}^1 \cup \mathcal{A}^2 \cup ... \cup \mathcal{A}^{t-1} - \mathcal{A}^t, \\
p_{\mathcal{A}^t_{out}} &= p_{\mathcal{A}^t_{in}}^{tan(\frac{\pi\Gamma^t}{2})} \cdot (1 - p_{in}^{\mathcal{F}^t})^{1 - tan(\frac{\pi\Gamma^t}{2})}, \\
p_{a_{out}} &= p_{a_{in}} \cdot \frac{p_{\mathcal{A}^t_{out}}}{p_{\mathcal{A}^t_{in}}}, \quad a \in \mathcal{A}^t, \\
p_{n_{out}} &= p_{n_{in}} \cdot (1 + \frac{p_{\mathcal{A}^t_{in}} - p_{\mathcal{A}^t_{out}}}{p_{\backslash(\mathcal{A}^t \cup \mathcal{F}^t)_{in}}}), \quad n \notin \mathcal{A}^t \cup \mathcal{F}^t,
\end{aligned}
\tag{4}
$$

where $\Gamma^t$ is the control strength for $\mathcal{A}^t$, and $\mathcal{F}^t$ is the frozen set, which consists of attribute-related tokens modified in previous turns but not in $\mathcal{A}^t$. As with Eq. 3, the sum of the probabilities of all tokens after modification remains 1. The main difference with Eq. 3 is that Eq. 4 introduces **the frozen set $\mathcal{F}^t$, in which the probabilities of tokens do not change in the $t$-th turn**.

### 3.3 CONTROLLABLE ATTRIBUTES

Through empirical examination and observation, we defined the following six controllable attributes at the token level (attribute-related tokens), which are used in later experiments.

**Sentence Length**    This is an objective attribute of text that is highly correlated with sentence enders (e.g., full stops, question marks, and exclamation marks). Increasing the probability of sentence enders makes the average sentence length shorter, and vice versa.

**Perfect Ending**    A dynamic $\Gamma$ tuning strategy of controllable SENTENCE LENGTH. Once the number of words generated exceeds a threshold, the value of $\Gamma$ is linearly decreasing when there are more words generated. As more words are generated, the more likely the model is to generate sentence enders like full stops. This ensures that the generated text does not stop mid-sentence.

**Topic Relevance**    Words that are most related to the user-given topic word (calculated by the cosine similarity of word embedding of GPT2) are selected to control the topic relevance of the generated text. In our implementation, we select the top 100 tokens.

**Sentiment Polarity**    We manually pre-build two word lists ($\approx$1K words) from the web with negative and positive words, respectively.

**Repetition Decreasing**    Similar to penalized sampling (Keskar et al. 2019), it prevents degeneration by decreasing the probabilities of tokens that have been generated recently.

**Relatedness Increasing**    A dynamic topic words tuning strategy of TOPIC RELEVANCE. Generating more coherent text by increasing the probabilities of words that are most related to recently generated nouns labelled by NLTK (Bird & Loper 2004).

The above is not an exhaustive list of attributes that can be achieved by GAMMA SAMPLING. All attributes that can be defined at the token level can be controlled by our method.

## 4    EXPERIMENTS

We conducted several experiments to evaluate the performance of GAMMA SAMPLING. Sec. 3.1 describes the metrics used in the experiment. Sec. 3.2 verifies the fine-grained control of GAMMA SAMPLING and the impact of its over-control strength through controllable sentence length. Sec. 3.3 evaluates the effectiveness of GAMMA SAMPLING in controlling topic and sentiment polarity through comparative studies.

### 4.1    METRICS

We evaluate the generated texts in four aspects: fluency, diversity, attribute relevance, and overall quality. The first three are evaluated automatically, while the overall quality is rated by MTurkers (human annotators from MTurk). To prevent keywords in the prompts from impacting these metrics, all texts used for the experiments removed the prompts, but retained the prefixes.

**Fluency**    The fluency of generated text is measured by GPT2 in different sizes based on perplexity (Vinyals & Le 2015). A higher perplexity (**PPL**) means that it is less likely that GPT2 will generate such text. It is important to note that a lower PPL is not always better, as texts with degeneration usually have an extremely low PPL, but their quality is generally considered to be very poor.

**Diversity**    DIST-1, DIST-2, and DIST-3 scores (Li et al. 2016) are used to evaluate the diversity of generated samples. A higher value of **DIST-N** means a higher proportion of distinct 1-2-3-grams in the generated text.

**Attribute Relevance**    We focused on evaluating three types of controllable generation: SENTENCE LENGTH, TOPIC RELEVANCE and SENTIMENT POLARITY. The average sentence length (**ASL**) measures the average number of words per sentence. The latter two are evaluated using two metrics based on external classifiers[12]: external classifier accuracy (**ECA**) and external classifier confidence (**ECC**). The higher the ECA and ECC, the more salient the external classifier considers the generated text to hold a certain topic/sentiment.

**Overall Quality**    We designed scoring criteria, TOEFL Writing Rubrics for Machine-generated Text (**T4MT**), to evaluate the overall quality of the machine-generated text. It is based on TOEFL Independent Writing Rubrics[3], a standardized evaluation of the English writing ability of non-native speakers accepted by more than 11,000 universities and other institutions. Given the characteristics of machine-generated text, we additionally set **whether the text itself is finished or contains factual errors does not affect the scoring, while extensive repetition, complete off-topic or obvious common sense errors will result in a low score**. MTurkers are asked to rate each generated text on a scale of 0 to 5 (nonsense to advanced) depending on its quality and whether it fits the given topic or sentiment. Further details of T4MT can be found in Appendix A.

---

[1] https://huggingface.co/facebook/bart-large-mnli
[2] https://huggingface.co/cardiffnlp/twitter-roberta-base-sentiment-latest
[3] https://www.ets.org/content/dam/ets-org/pdfs/toefl/toefl-ibt-writing-rubrics.pdf

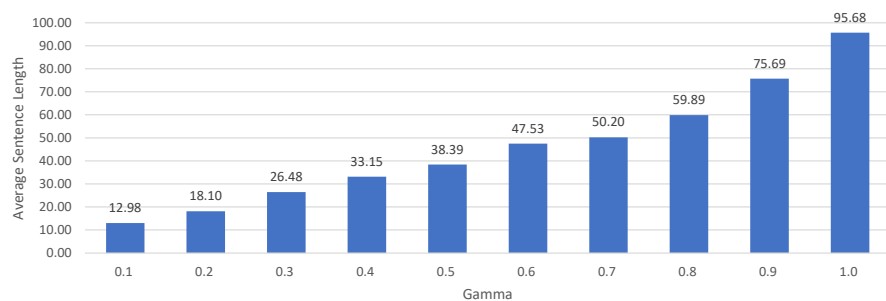

Figure 4: Average Sentence Length (ASL) in various settings of Γ (Gamma).

Table 1: Results of controllable sentence length in various settings of Γ and top-$p$.

| top-$p$, Γ | PPL-Small↓ | DIST-1↑ | DIST-2↑ | DIST-3↑ | ASL |
|---|---|---|---|---|---|
| **1.0, 0.9** | 64.65 | 62.90 | 78.02 | 80.71 | 75.69 |
| **0.9, 0.9** | 21.33 | 59.72 | 76.85 | 80.68 | 59.21 |
| **0.8, 0.9** | 12.49 | 55.24 | 72.32 | 76.18 | 53.92 |
| **1.0, 0.5** | 56.95 | 62.27 | 78.22 | 81.48 | 38.39 |
| **0.9, 0.5** | 20.42 | 60.98 | 78.34 | 81.94 | 31.54 |
| **0.8, 0.5** | 15.14 | 58.11 | 76.20 | 80.56 | 25.33 |
| **1.0, 0.1** | 6844.10 | 64.40 | 69.60 | 68.37 | 12.98 |
| **0.9, 0.1** | 48.75 | 51.59 | 61.98 | 64.19 | 15.25 |
| **0.8, 0.1** | 16.53 | 48.08 | 61.00 | 64.10 | 10.84 |

## 4.2 CONTROLLABLE SENTENCE LENGTH

The model used in this experiment was GPT2-Small (117M). All generated samples start with the same prefix `The issue focused` and the model is asked to generate the following 100 tokens. To ensure that each sample has at least one sentence ender, we enabled PERFECT ENDING, so that Γ decreases linearly to 0 (when generating the 100th token) as the number of tokens generated increases, starting at the 80th token. Some typical generation samples of controllable sentence length can be found in Appendix B.

We first investigated the fine-grained control of GAMMA SAMPLING, generating 100 samples with random seeds from 0 to 99 at each different value of Γ, for a total of 1000 samples. As shown in Fig. 4, there is a significant increase in ASL with increasing values of Γ, from an average of 12.98 words per sentence (Γ = 0.1) to an average of 95.68 words per sentence (Γ = 1.0), while the average number of words per sentence generated by GPT2-Small is 38.39 (Γ = 0.5). It is worth noting that ASL shows a linear increase when Γ grows, suggesting that the probability scaling of attribute-related tokens has a non-linear impact on the corresponding attribute. In addition, if PERFECT ENDING is not turned on, a sentence will never end when Γ = 1.0.

To examine the risk of the over-control strength, we then selected three representative Γ (0.1, 0.5, and 0.9) and used nucleus sampling with top-$p$ from 0.8 to 1.0. Nucleus sampling is used before GAMMA SAMPLING (as was the case for the rest of the experiments in this paper). In each of these nine settings, 100 samples were generated for evaluation. As previously discussed, arbitrarily increasing the probability of controlled tokens significantly reduced the quality of the generated text (see Table 1). When top-$p$ = 1.0 and Γ = 0.1, the PPL of the generated samples reaches 6844.1, which means that it is almost impossible for GPT2-Small to generate such text. However, by setting a lower top-$p$, the impact of GAMMA SAMPLING on the quality of the generation can be greatly alleviated. When top-$p$ = 0.8, the PPL drops dramatically to 16.53 even with still very extreme Γ = 0.1. In contrast, the impact of drastically reducing the probability of controlled tokens (Γ = 0.9) is much smaller.

In summary, by scaling the probabilities of attribute-related tokens, GAMMA SAMPLING can significantly manipulate the corresponding attributes of generated samples. To alleviate the negative impact of the over-control strength, pre-sampling (e.g., nucleus sampling) is needed.

Table 2: Main results for comparing all controlling methods applied nucleus sampling (top-$p = 0.8$). All the best results are bolded in red. Due to space constraints, the GPT2, GS, and GSM reported here are averages of the results on all sizes of GPT2. In addition, the PPL here is the average of the PPLs calculated from all sizes of GPT2. Detailed results are presented in Appendix D.

| Method | PPL↓ | DIST-1↑ | DIST-2↑ | DIST-3↑ | ECA↑ | ECC↑ | T4MT↑ |
|---|---|---|---|---|---|---|---|
| GPT2 | 11.71 | 44.85 | 59.49 | 64.19 | 58.56 | 51.39 | 2.71±0.96 |
| GS | **8.24** | 36.52 | 49.81 | 54.60 | 56.79 | 51.46 | 2.76±1.01 |
| GSM | 23.74 | **78.10** | **93.72** | 95.17 | **60.73** | **54.03** | **3.35±0.77** |
| GPT2-FT | 27.61 | 53.90 | 71.96 | 76.88 | 48.59 | 44.53 | 3.08±0.93 |
| PPLM-BCR | 11.01 | 60.79 | 85.80 | 90.54 | 43.50 | 40.08 | 3.18±0.83 |
| K2T | 20.24 | 69.93 | 93.59 | **96.32** | 39.50 | 38.08 | 3.15±0.80 |
| CTRL | 13.37 | 46.74 | 65.74 | 71.54 | 51.67 | 45.38 | 3.12±0.89 |

Table 3: Comparison of different methods. PPLM, K2T, and GAMMA SAMPLING do not require prompts as they do not introduce attribute-related information at the input. The generation time of K2T increases linearly with the number of keywords. Since the time cost of GAMMA SAMPLING is insensitive to the number of controlled tokens, the generation times of GS and GSM are basically identical. GAMMA SAMPLING not only adds no parameters, but is computationally efficient.

| Method | Generation Times (GPU sec/sample) | Prompt for topic | Prompt for sentiment | Number of parameters |
|---|---|---|---|---|
| GPT2-Small | 2.89 | topic: [TOPIC] article: | topic: [SENTIMENT] reviews: | 117M |
| GPT2-Medium | 3.64 | topic: [TOPIC] article: | topic: [SENTIMENT] reviews: | 345M |
| GPT2-Large | 5.66 | topic: [TOPIC] article: | topic: [SENTIMENT] reviews: | 774M |
| GPT2-XL | 7.93 | topic: [TOPIC] article: | topic: [SENTIMENT] reviews: | 1.6B |
| Fine-tuned GPT2 | 4.02 | topic: [TOPIC] article: | topic: [SENTIMENT] reviews: | 345M |
| CTRL | 17.28 | [TOPIC] Text: | Reviews Rating: 1.0 (or 5.0) | 1.6B |
| PPLM-BoW (topic) | 593.24 | – | – | 345M+0 |
| PPLM-Discrim (sentiment) | 1863.43 | – | – | 345M+≈1K |
| K2T | 10.45/keyword | – | – | 774M+0 |
| Gamma Sampling | +2.99 | – | – | +0 |

## 4.3 CONTROLLABLE TOPIC AND SENTIMENT

As shown in Table 2, for the controllable topic and sentiment, we include the following methods.

**GPT2** (Radford et al. 2019)   Vanilla GPT2 models come in four sizes: Small (117M), Medium (345M), Large (774M), and XL (1.6B). No other controlled generation techniques were used, other than the introduction of attribute-related information via prompt.

**GS**   A baseline GAMMA SAMPLING that only enables TOPIC RELEVANCE or SENTIMENT POLARITY to control GPT2 in four sizes, i.e., only increases the probability of attribute-related tokens ($\Gamma = 0.1$).

**GSM**   Besides TOPIC RELEVANCE or SENTIMENT POLARITY, it enables multiple controllable attributes, i.e., REPETITION DECREASING ($\Gamma = 0.9$) and RELATEDNESS INCREASING ($\Gamma = 0.3$).

**GPT2-FT**[4]   GPT2-Medium fine-tuned on a large news corpus for topic-controllable generation. Zero-shot generation works pretty well as long as the topic is a single word and not too specific.

**PPLM-BCR** (Dathathri et al. 2020)   It guides GPT2-Medium and has two different schemes (see Table 3) for the controllable topic and sentiment, respectively. In our experiment, both of them update the latent representations, generate 10 samples, and choose the best sample based on the log-likelihood. In the main results, they are referred to together as PPLM-BCR.

---

[4] https://huggingface.co/ktrapeznikov/gpt2-medium-topic-news

**K2T** (Pascual et al. 2021) A guided decoding method modifies the score function to encourage GPT2-Large to generate more words that are semantically similar to keywords in a given set (i.e., attribute-related tokens). To ensure fairness, K2T uses the same keyword set as GAMMA SAMPLING. However, its time cost increases linearly with the number of controlled tokens, and the generation time is unacceptable when using all keywords from SENTIMENT POLARITY ($\approx$10,000 sec/sample). Therefore, we randomly selected 5 of the attribute-related tokens used by GAMMA SAMPLING as keywords for each K2T-generated sample, as is the default setting for this method. In our experiments, all hyperparameters of KT2 are set to default values.

**CTRL** (Keskar et al. 2019) A 1.6 billion-parameter conditional transformer language model trained from scratch to condition control codes (i.e., prompts).

Following the experiments of PPLM (Dathathri et al. 2020), three topics (COMPUTERS, LEGAL and SCIENCE) and two sentiment polarities (NEGATIVE and POSITIVE) are tested in our experiments. To avoid ethical issues, we removed some of the topic generation (e.g., POLITICS and RELIGION). All models are asked to generate 100 essays for each task. Given the prefix The issue focused for topics and The movie for sentiments, combined with their respective prompts, models need to continue to write the following 100 words. In addition, we applied nucleus sampling with top-$p = 0.8$ for all methods. For automatic evaluation, we get $16 \text{ methods} \times 5 \text{ tasks} \times 100 \text{ generations} = 8000$ samples in total, and for human evaluation, we selected the first 30 generations for each model per task, in total $16 \text{ methods} \times 5 \text{ tasks} \times 30 \text{ generations} = 2400$ samples. We received a total of 7285 valid ratings, with each sample being reviewed by at least 3 MTurkers. We provide some generation samples of all methods in Appendix C.

Table 2 shows that GS-generated texts have a lower PPL than GPT2-generated ones, but are the worst in terms of diversity metrics (DIST-N), while there is no significant difference in attribute relevance metrics (ECA and ECC) and the overall quality metric (T4MT). After manual examination, we found that the GS-generated text showed more degeneration compared to the GPT2-generated ones, i.e., was more likely to repeatedly generate words with similar semantics to the topic words. Therefore, using TOPIC RELEVANCE or SENTIMENT POLARITY alone did not achieve good performance on the controllable text generation task. In contrast, GSM-generated texts show a very high degree of diversity while maintaining high attribute relevance, and in terms of the standard deviation of T4MT, the quality of the text generated by GSM is the most stable of all methods. Although GSM sacrifices a certain degree of fluency (PPL), it does not reach an unacceptable level, according to T4MT. These show that using REPETITION DECREASING and RELATEDNESS INCREASING together can significantly improve the quality of the generated text without going off-topic.

Although, in terms of T4MT, most of the methods fell within the uncertainty boundaries of each other, we performed independent samples $t$-tests, and found statistically significant differences in the GSM results compared to all the other methods (except PPLM-BCR), i.e. $p$-value $< 0.05$. For PPLM-BCR, the $p$-value is 0.051. While we cannot claim, strictly speaking, that the quality of GSM-generated texts is significantly better than that of PPLM-BCR, the generation time of PPLM-BCR is at least $100\times$ longer than that of GSM, as shown in Table 3.

K2T, which is also a guided decoding method, ECA and ECC show that it generates more off-topic text compared to GAMMA SAMPLING. We suggest that this is due to the fact that K2T defaults to a weaker control strength to avoid off-distribution caused by over-control strength, a conclusion that can also be drawn from their paper (Pascual et al. 2021). GAMMA SAMPLING, on the other hand, is affected negligibly by this issue, just as we used a fairly strong control strength in our experiments.

## 5 CONCLUSIONS

We presented a method for fine-grained controlling of LMs by defining attributes at the token level, which can be pre-built or selected by the LMs themselves. We demonstrated the fine-grained control of GAMMA SAMPLING in our experiments and showed that GAMMA SAMPLING-steered GPT2 generally outperformed all baselines on both objective and subjective metrics while maintaining a fast generation speed. Compared to K2T, it is not only more computationally efficient, but can also avoid off-distribution caused by over-control strength via using nucleus sampling as pre-sampling. As GAMMA SAMPLING is a training-free method, it can be easily used for any LM, and shows promise in steering LMs toward more efficient, user-friendly, and data-free controllable generations.

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

## APPENDIX A

**Please read the "General Rules" below carefully to ensure that you understand the scoring criteria. More details are given in the "Writing Rubrics" (in full instructions). If your score is too far away from the average of other annotators (difference in absolute value greater than 1), it will be rejected.**

**Notice:** The contents of these essays are not necessarily factual so do not believe exactly what you read. If an essay contains a link to a web page, do not access it in your browser.

**General Rules:**

- Each essay needs to be rated on a scale of 0 to 5 depending on its quality and whether they fit the given topic
- Most of the essays are around 100 words in length, and most of them stop mid-sentence (essays are not finished). It is common and does not affect the scoring
- If an essay is completely off-topic, or included something that clearly contradicts common sense, only a maximum of 1 mark can be scored
- If an essay contains factual errors (e.g., South African anti-Apartheid leader Nelson Mandela dying in prison in the 1980s) but not common sense errors (e.g., the sun rises in the west), it does not affect the scoring

**Writing Rubrics[5]:**

| Score | Level | TASK DESCRIPTION |
|---|---|---|
| 5 | Advanced | **An essay at this level largely accomplishes all of the following:**
✓ Effectively addresses the topic and task
✓ Is well organized and well developed, using clearly appropriate explanations, exemplifications and/or details
✓ Displays unity, progression and coherence
✓ Displays consistent facility in the use of language, demonstrating syntactic variety, appropriate word choice and idiomaticity, though it may have minor lexical or grammatical errors
✓ Does not contain any obvious errors of common sense |
| 4 | High-Intermediate | **An essay at this level largely accomplishes all of the following:**
✓ Addresses the topic and task well, though some points may not be fully elaborated
✓ Is generally well organized and well developed, using appropriate and sufficient explanations, exemplifications and/or details
✓ Displays unity, progression and coherence, though it may contain occasional redundancy, digression, or unclear connections
✓ Displays facility in the use of language, demonstrating syntactic variety and range of vocabulary, though it will probably have occasional noticeable minor errors in structure, word form or use of idiomatic language that do not interfere with meaning
✓ May occasionally contain common sense errors but do not affect the presentation |
| 3 | Low-Intermediate | **An essay at this level is marked by one or more of the following:**
✓ Addresses the topic and task using somewhat developed explanations, exemplifications and/or details
✓ Displays unity, progression and coherence, though connection of ideas may be occasionally obscured
✓ May demonstrate inconsistent facility in sentence formation and word choice that may result in lack of clarity and occasionally obscure meaning
✓ May display accurate but limited range of syntactic structures and vocabulary
✓ May contain common sense errors rendering the presentation dubious |
| 2 | Basic | **An essay at this level may reveal one or more of the following weaknesses:**
✓ Limited development in response to the topic and task
✓ Inadequate organization or connection of ideas
✓ Inappropriate or insufficient exemplifications, explanations or details to support or illustrate generalizations in response to the task
✓ A noticeably inappropriate choice of words or word forms
✓ An accumulation of errors in sentence structure and/or usage
✓ May contain obvious common sense errors that make the presentation obscured |
| 1 | Below Basic | **An essay at this level is seriously flawed by one or more of the following weaknesses:**
✓ Serious disorganization or underdevelopment
✓ Little or no detail, or irrelevant specifics, or questionable responsiveness to the task
✓ Serious and frequent errors in sentence structure or usage
✓ May contain a lot of sentence-level repetition
✓ May contain serious common sense errors rendering the presentation difficult to understand |
| 0 | Nonsense | **An essay at this level** is a complete mess consisting of random characters and/or a great amount of word-level or phrase-level repetition. |

---

[5]mainly based on TOEFL iBT Independent Writing Rubrics
https://www.ets.org/content/dam/ets-org/pdfs/toefl/toefl-ibt-writing-rubrics.pdf

APPENDIX B

Examples of SENTENCE LENGTH generated by GPT2-Small in various settings of $\Gamma$. The underlined prefix is what the LM is conditioned on to generate the text.

$\Gamma = 0.1$    The issue focused on today's visualizing of America. At one point? Will our massive commercialisation interest allow us to avoid such digital environments? For much of that space? Is there any hope? I really thought we'd?" he told your Daily Mail. "For most other places in the world? Don't say it for real. Don't put people." Addressing another episode of Call Me By Your Name.

$\Gamma = 0.2$    The issue focused on practice and implementation of criminal responsibility for Caplindale's planned smoking ban. It came after court orders told officials that Smokers Protection Society could always appeal the ban. CITES held about 60 meetings with department officials in April. An employee said there were several meetings where seven states had ordered Smokers Protection Society to copy booklets from Smokers Protection Society. These conflict with Smokers Protection Society's views of enforcement.

$\Gamma = 0.3$    The issue focused on student contracts and debt at Wal-Mart. As such, members have essentially zero input into whether government agencies should collect student contracts or stop collecting them. This led to big gains for Wal-Mart by participating in workers' collective bargaining (WCT). But now we're seeing evidence that Wal-Mart only receives federal revenue from student contracts and won't pick up some of it from store employees. It also produced bad results for minimum wage increases.

$\Gamma = 0.4$    The issue focused on Muslim rape laws and forced marriages in 1999. Mr Khaled told local magistrates that after five years of preparation for his trial in 2004, he began feeling uncomfortable about going to court at all. He received several letters from male guests telling him he could not enter his date's house under certain circumstances: his partner would only return when they return from work; his partner would return when they leave as well; or if they tried to enter his house with him.

$\Gamma = 0.5$    The issue focused on economy and Social Security was raised when MacCallum proposed 2010 increases in taxes on college tuition and connected wages to reduce benefits for Americans who received college degrees to increase income among workers who received higher education. For example, President Obama proposed finding $32 billion in zero-income tax credits for individuals under age 65 with college degrees (who currently receive home ownership systems from low-income households). These taxes would continue to fall under current tax laws and leave everyone without enough money to pay for."

$\Gamma = 0.6$    The issue focused on trustful providers for Canadian consumers and helped shape how Canada named its internet policies and regulations in recent years. Disclosure: Canadian Federation of Independent Business Inc. policy gives university perspective on internet privacy from Keening There have been roundups in various meetings over the past month with Canadian universities that offer options for cloud hosting but those solutions don't require books or accessible internet access. Part of that includes experts able to blog about product launch events and deliver smaller updated versions of works.

$\Gamma = 0.7$    The issue focused on trust between banks and consumers that became hot under President Donald Trump's administration. This included policies such as limits on Wall Street bailouts and restrictions on immigration from various African countries. It also focused on climate change and stated that financial institutions should consider reducing costs as well as requiring investment planning for increased risks and duration of risks to businesses in order to reduce stock prices. The strategy involved focusing on environmental concerns as well as assets such as servers and loan guarantees.

$\Gamma = 0.8$    The issue focused on trust between banks and consumers for increased transparency and transparency from consumer agents and rated agencies that sell digital currency which require banks to label different consumers "qualified trust agents or rated agencies," according to NIST officials. Speculation over whether Bitcoin could fall through this winter now continues to haunt Bitcoin investors as policy makers fix issues through legislation that require banks to label customers "qualified trust agents or rated agencies," according to. And around the world a ban on Bitcoin units like trading accounts or dealing in.

$\Gamma = 0.9$    The issue focused on trust between banks and consumers with regard to EU controls on retail banking while animal welfare theme took center stage at an EU summit on animal welfare this week in Brussels. Under EU financial controls on retail banking the animal welfare theme had little impact on climate change global warming policy," according to recently published comments from Food say systems using river stocks have lower chances of sustained desertification than systems using sustainable streams with sufficient nutrients for mild drought and desertification.

$\Gamma = 1.0$    The issue focused on trust between banks and consumers with regard to EU controls on retail banking while animal welfare theme took center stage at an EU summit on animal welfare this week in Brussels (March 25-26), according to French expert Reynaud Naisonge Dengeau of Merck & Co., which supports EU consumer protection measures (pdf), Google Cell or Android OS X version 7 or higher (encryption security another factor there), net neutrality taking center stage at an EU summit on animal welfare this week in Brussels.

## APPENDIX C

### TOPIC: COMPUTERS

The following examples are generated by various methods. The topic here is `COMPUTERS`. The prompt is coloured red, and combined with the underlined prefix is what the LM is conditioned on to generate the text. PPLM-BoW, K2T, and GAMMA SAMPLING (GS and GSM) do not require any prompt.

**GPT2-Small**   topic: computers article: The issue focused on natural selection on physical processes and physical processes are related to computer evolution: Part 2 A theory of evolution The theory of evolution claims that nature has evolved from man and man has evolved from plant and animal life. These processes can be understood as selection for natural selection and predation on animals. Over time, natural selection will eventually produce human beings that have higher intelligence and higher survival rates. Some scientists believe that humans will eventually surpass animals and become intelligent as human beings. Some scientists believe that humans will eventually

**GPT2-Medium**   topic: computers article: The issue focused on how more recent versions of Windows no longer support x86 systems. - learn more on The issue focused on how more recent versions of Windows no longer support x86 systems. - learn more on http://www.howtogeek.com/how-to-configure-one-workstation-for-the-next-generation-of-your-computer/ How to write a command line tool in Ruby What is the difference between an application and a command line

**GPT2-Large**   topic: computers article: The issue focused on Amiga computers is their capacity to store data and perform tasks in parallel. How many servers do you need to support your network of clients? How many servers do you need to support your network of clients? How many servers do you need to support your network of clients? No matter how good your hardware is, if you don't have enough servers to serve your clients, you'll never get them all working. No matter how good your hardware is, if you don't have enough servers to serve

**GPT2-XL**   topic: computers article: The issue focused on net neutrality and whether or not certain organizations like Facebook or Google should be allowed to charge extra fees for faster internet connection. As this article explained, many companies like Facebook and Google use so called "zero rating" plans that provide special treatment for certain apps or websites. These plans give users access to certain apps or websites at no additional cost. However, many companies like Facebook and Google don't actually provide any content for free. Instead, they offer services for free (like Facebook Connect, which lets

**GS-Small**   The issue focused on Windows 10 hardware devices and devices running Windows 10 Home Edition. Windows 10 Home Edition devices also included Windows 10 Mobile devices. Windows 10 Mobile devices include devices running Windows 10 Mobile devices that use Windows 10 Mobile devices in Windows 10 Mobile devices. Windows 10 Mobile devices also include devices running Windows 10 Mobile devices that use Windows 10 Mobile devices in Windows 10 Mobile devices. Windows 10 Mobile devices also include devices running Windows 10 Mobile devices that use Windows 10 Mobile devices in Windows 10 Mobile devices. Windows 10 Mobile devices

**GS-Medium**   The issue focused on technology available to internet companies to conduct internet surveillance. With technology available to internet companies to conduct internet surveillance, internet companies can conduct internet surveillance at will. This technology does not require internet service providers to obtain an internet search warrant. As technology enables internet companies to conduct internet surveillance, internet companies can conduct internet surveillance at will. This technology does not require internet service providers to obtain an internet search warrant. To conduct internet surveillance, internet companies must obtain internet content information (IPI) from internet

**GS-Large**   The issue focused on technology — software that monitors cars and sensors that detect cars on the road — as well as software that monitors cars and sensors that detect cars on the road. The software monitors cars for signs of failure and sensors for injuries. The hardware monitors cars for symptoms and sensors for injuries. "You need hardware and software that do this without software," says Russel Johnsen, who leads NHTSA's Mobile Electronics Electronics Electronics Electronics Electronics Electronics Electronics Electronics Electronics Electronics Electronics Electronics Electronics Electronics Electronics Electronics Electronics Electronics Electronics Electronics

**GS-XL**   The issue focused on French electronic hardware manufacturer Logitech's gear and software software and Windows PC hardware and software for tracking devices. Microsoft Windows PC hardware and software software software software software software software software software software software software software software software software software software software software software software software software software software software software software software software software software software software software software software software software software software software software software software software software software software software software software software software software software software software software software software software software software software software software software software software software software software software software software software

**GSM-Small**    The issue focused on Windows 10 hardware devices and Internet Explorer software. Microsoft engineers created the framework that monitors applications from desktop PCs to tablets, smartphones and laptops using Wi-Fi networks. The goals are similar to those for websites or databases in browsers: servers take care of communication between computers; sensors measure computer activity and traffic volume; and algorithms automatically identify machines as belonging to one particular machine group. Even though Windows 10 technology has yet to become widely adopted by developers, hackers who believe they can steal systems will often

**GSM-Medium**    The issue focused on Facebook's announcement that it would begin requiring internet companies to store user data for two years in servers located outside the United States. The technology company said it had received thousands of consumer complaints about its practices. Mr. Pewson was critical of Facebook's application of digital advertising techniques, saying they too often presented websites with misleading ads and automated messages about friends' activities. He called them "deceptive digital technologies."From Microsoft Windows systems to smartphones, computers are becoming devices that perform computer tasks

**GSM-Large**    The issue focused on technology issues that face digital technologies and internet freedom. In particular, cyber threats like hackers targeting websites are believed to have reached levels never seen before in history. And internet freedom was another central theme of the conference. Notably, cyber threat cyber terrorism has become a hot topic over the past few years. Yet cyber security remains extremely complicated and technological solutions remain elusive. Security experts fear cyber terrorists could deploy advanced offensive cyber capabilities against computer systems or computers systems can be hacked into without cyber capability to

**GSM-XL**    The issue focused on technology that Microsoft Windows devices could use to store digital files. The software giant had shown off the technology as part of an Internet Explorer browser update in April 2012. But critics say Microsoft did not implement it properly. On Tuesday, Microsoft announced a deal with hackers who broke into Apple's iCloud servers and stolen nearly 60 million photos from iPhones and iPads. Microsoft said it would give hackers access to those images so they can study them for malware.Deuteronomy 17:9-20 New

**GPT2-FT**    topic: computers article: The issue focused on both ways in which technology can make use of already existing assets in order to reach new markets. One issue involved research into whether services such as chat apps could work well with existing infrastructure. While some researchers believe this could work, others believe it would not be practical to offer such services on the web. Some suggested that having no mobile internet service would make it easier for criminals to crack security systems. Google said it would provide access to two Google Apps services that help businesses save money on cloud computing and

**PPLM-BoW**    The issue focused on the way police handled and recorded audio recordings, according to the ACLU's report, which also found that police officers sometimes use cell-site technology to record and analyze data on the internet. The ACLU report, titled "The Rise of Cell Stealing by Mobile Data Devices," was released today at the annual meeting of the American Civil Liberties Union International. The report, which examines the privacy implications of mobile surveillance technologies, also looks at other privacy-related issues, including privacy of communications data in

**K2T**    The issue focused primarily on the wording of a section of the state constitution which criminalises the organisation of prostitution, but also included a provision criminalising the group of prostitutes who work separately in houses of prostitution. If women buy and sell sex and the group are classified as sex workers, they can be charged with offences of providing or facilitating prostitution and to encourage them to enter the trade. However, the state did not include such an explicit provision in its updated prostitution act that will come into force next year which,

**CTRL**    Technologies Text: The issue focused on making sure that government agents could get access to text messages if they wanted to. First off, there are several requirements for getting access to text messages. Most importantly, you need to know that a message is considered protected by law in your jurisdiction. Most messages will also need to be verified by a central database (Microsoft Outlook or WhatsApp). Second off, there are different levels of interception (example: voice, video, etc.). Most messages will need to be monitored by some sort of monitoring tool (example: PGP)

Topic: Legal

The following examples are generated by various methods. The topic here is LEGAL. The prompt is coloured red, and combined with the underlined prefix is what the LM is conditioned on to generate the text. PPLM-BoW, K2T, and Gamma Sampling (GS and GSM) do not require any prompt.

**GPT2-Small**    topic: legal article: The issue focused on issues related to gender transition and gender expression in science and technology; following this led to these articles which appeared in daily newspapers and educational outlets; found support for information about gender transition and gender expression in science and

technology; created article which describes research into gender transition and gender expression in science and technology; discovered recent scientific literature which supports male and female transgender individuals in science and technology; stated that science and technology should create evidence that gender transition is better than gender expression; suggested that gender transition can be indicated

**GPT2-Medium**   topic: legal article: The issue focused on whether current criminal legislation made any difference to how users used search engines in Canada. — Source: Special to The Globe and Mail Download article (PDF, 394KB) — Full text (PDF, 382KB) — Audio (PDF, 109KB) — PDF View original text Copyright © 2000 The Globe and Mail Back to Top Report by Jay Hartman and Robin Brandis of Law Centre for Constitutional Freedoms on Project CRIME and charges brought against

**GPT2-Large**   topic: legal article: The issue focused on not only your written document, but also your speech: You could then question them for other reasons. You could ask them about their behavior in the past. You could ask them about their behavior in the future. You could ask them about their behavior in the past. You could ask them about their behavior in the future. And they could tell you any number of things: The policy was written by an unknown person. The policy was written by an unknown person. The

**GPT2-XL**   topic: legal article: The issue focused on being able to download content from online sources like Xbox Live and PlayStation Network and having it installed on an Xbox 360 without having to buy an expensive digital copy of the game. As one example, many owners of games like Halo 3 have paid more than $70 for the digital copy of the game. With this article, we'll discuss what has happened with games like Halo 3 and why some developers have moved away from digital downloads in favor of selling physical copies of their games. This article will provide some

**GS-Small**   The issue focused on potential legislation that would legalize medical marijuana and medical marijuana businesses. The legislation would legalize medical marijuana businesses and medical marijuana businesses would have to prove they would get legal medical marijuana coverage from federal law. These factors could include medical marijuana businesses filing lawsuits or filing lawsuits against medical marijuana businesses that don't provide coverage for medical marijuana business tax credits. However, medical marijuana businesses would need to prove they would meet federal rules for coverage that include medical marijuana businesses filing lawsuits or filing lawsuits against medical marijuana

**GS-Medium**   The issue focused on plans for medical marijuana legislation in Colorado and Washington. Lawmakers have moved toward decriminalizing possession of small amounts of marijuana and legalizing recreational use of the drug. However, opposition from medical marijuana advocates has pushed legislators to prohibit medical marijuana programs from receiving federal funds. This legislation would prohibit federal funds from going to medical marijuana programs that prohibit medical marijuana patients from receiving federal funds. Dr. Lenora Jackson, medical director of an Oakland medical marijuana dispensary, told legislators this legislation would prohibit medical

**GS-Large**   The issue focused on federal standards for enforcement of laws intended to address illegal immigration. The legislation would bar federal officials from suing states for compliance with laws related to immigration enforcement. According to federal data, illegal immigrants make up only about 7 percent of the federal workforce. Law enforcement officers make up about 80 percent of federal immigration enforcement officers. Even if federal enforcement officers sue states for compliance with laws related to illegal immigration, federal courts would likely decide that federal authorities do not have standing to sue because federal authorities

**GS-XL**   The issue focused on Congress's creation of the Supreme Court seat for Justice Anthony Kennedy, who will step down from the court this summer. Lawmakers used the courts to legislate political priorities and secure additional Supreme Court appointments. Lawmakers say they want to avoid political gridlock and send a message to President Obama that they want him to fill the seat. While nearly every justice said they support the legislation, Justice Scalia also suggested that political considerations should not influence decisions that affect public policy. "If you

**GSM-Small**   The issue focused on China's financial regulatory enforcement and management of criminal enterprises. According to court documents, prosecutors alleged that federal authorities prosecuted illegal Chinese firm Law Criminal Legal Enterprise Group (CCLGE) for engaging in illicit activities such as trading unlawful currency or goods at legal tender. Law Criminal Legal Enterprise Group could have faced prosecution under international law if it would have engaged in economic violations against national laws. China also reportedly seized tax assets worth $18 billion from lawyers involved in filing lawsuits challenging government corruption charges

**GSM-Medium**   The issue focused on laws requiring criminal prosecutions for illegal copyright infringement. Lawmakers wanted legislation that would prohibit prosecutors from prosecuting users who illegally downloaded songs or movies. The bill passed in 2011 and law enforcement officials saw it as the legal equivalent of banning cocaine. While judges generally believed criminal prosecution was unconstitutional, courts gave lawyers rights to prosecute digital crimes under federal copyright laws. Lawyers say courts used copyright

laws to sue plaintiffs when they allegedly stole copies of copyrighted material without permission. Lawyers typically argued copyright lawsuits could help civil litigation

**GSM-Large**     The issue focused on whether women who had legal abortions were protected from liability for criminal prosecution in civil cases. In court, lawyers argued that federal law permitted abortion rights activists to sue medical providers and courts could not legally judge the legality of legal abortions. In court documents filed last month, attorneys for Planned Parenthood also noted that federal law allows lawsuits against medical facilities that provide abortions. The lawsuit said that laws criminalizing abortion rights supporters have left legal clinics without financial resources to cover medical bills. This week's

**GSM-XL**     The issue focused on recent legislation that requires political parties to keep government financial records public. Party officials had argued the law would limit political debate and civil rights. But Justice Antonin Scalia said it was constitutional for courts to apply laws "contrary to constitutional principles." He wrote that federal courts should make rulings based on legal precedents. In his dissent, Justice Samuel Alito called the court's ruling a "great judicial irony." "On one hand, Justice Scalia holds that laws must be

**GPT2-FT**     topic: legal article: The issue focused on by police after the deaths of three young men was how they came to end up on the streets of Birmingham.They were shot dead by two police officers on 16 October after they stopped and searched them for drugs in West Kirkham Street.Some officers may have known about the drugs but did not stop them and made no arrests.One officer has been suspended while an internal investigation is carried out. Police said they had received some allegations that police officers had stopped them for drugs on at least two occasions.

**PPLM-BoW**     The issue focused on the way police handled and captured suspects who had been charged with murder in cases, according to the suit. One defendant, identified in court papers as Jerald J. Williams, died of a heart attack, and the other defendant, identified in court papers as David R. Williams, died of a stroke. Prosecutors charged the two of them with first-degree murder. Prosecutors said that Williams had been convicted of murder after a police officer killed the man's girlfriend and then killed himself

**K2T**     The issue focused on a Secret Service officer who allegedly witnessed the officers' alleged actions and alerted officials that he was under investigation. Officers are still under investigation for possible administrative violations. ADVERTISEMENT In a letter sent to Senate Judiciary Committee Chairman Charles Grassley Charles (Chuck) Ernest GrassleyHow Flake came to secure Kavanaugh delay GOP leaders delay Kavanaugh confirmation for one-week FBI investigation Judiciary approves Kavanaugh, sending nomination to full Senate after Flake request MORE (R-Iowa) and House Oversight and Government Reform Committee Chairman

**CTRL**     Legal Text: The issue focused around my ex telling me I should try to open up a credit card at her store when it is open at another location. She does this about once a week. After about three weeks of this, I decided to ask her if she can open up a store right next to where she works. She asked me if I could open up a store right next to her store but she told me she didn't want me to open up a store right next to her store. She even told me not to open up a store right next

TOPIC: SCIENCE

The following examples are generated by various methods. The topic here is SCIENCE. The prompt is coloured red, and combined with the underlined prefix is what the LM is conditioned on to generate the text. PPLM-BoW, K2T, and GAMMA SAMPLING (GS and GSM) do not require any prompt.

**GPT2-Small**     topic: science article: The issue focused on conditions at most microalgae samples (submillimeter scale). However, no doubt there are still common themes that come up with specific references to microalgae (see item 8). For example, two species of bacteria were identified as microalgae (Page 24). These bacteria were identified by comparison with non-microalgae samples (Page 22). These bacteria were identified by comparison with microaerobes (Page 23). These bacteria were identified by comparison with microalgae samples (Page

**GPT2-Medium**     topic: science article: The issue focused on air pollution is already receiving attention by researchers for use in environmental protection efforts description: Air pollution has long been considered an environmental threat to human health and public health. Now, research shows that air pollution may lead to mental illness and suicide description: Life expectancy at birth has declined since its peak in the 1960s description: Labor force participation rate has declined since its peak in the 1960s description: Children living in poverty have lower educational attainment than children living in poverty

**GPT2-Large**     topic: science article: The issue focused on could get too expensive for research for things like genetic engineering to keep humans alive in the future. As one scientist told me, "You want to keep people alive for as long as possible. For some reason, we don't want to pay for what we're doing." It's no surprise that people would pay more for certain kinds of science that help them live longer. And they don't want to pay more for certain kinds of science that help them live longer. But if we're going to

**GPT2-XL**   topic: science article: The issue focused on two objects in space: an asteroid that had come close to Earth in 2005 and an asteroid that had passed within 16,000 miles (26,000 kilometers) of Earth in October 2012. This article featured data from NASA's Near Earth Object Observations Program (NEOWISE), which tracks asteroids and comets with ground-based telescopes. NEOWISE uses two telescopes to gather data on asteroid sizes and orbits. On Oct. 25, 2012, NEOWISE detected an asteroid about

**GS-Small**   The issue focused on climate change and climate change scientists have noted climate scientists have turned their attention to climate change as the answer to climate change. Scientists have suggested climate scientists will consider climate change more strongly if climate scientists research climate change as the answer to climate change. Scientists have suggested climate scientists will consider climate change more strongly if climate scientists research climate change as the answer to climate change. Scientists have suggested climate scientists will consider climate change more strongly if climate scientists research climate change as the answer to climate change. Scientists have suggested

**GS-Medium**   The issue focused on medical marijuana research and medical marijuana programs in Colorado and Washington. The Senate passed the medical marijuana measure after medical marijuana supporters successfully urged supporters to carry it over to the House. While medical marijuana research has received attention from researchers for many years, medical marijuana programs have received less attention from researchers and researchers have suggested medical marijuana programs could become less effective if medical marijuana research did not receive enough funding. According to House Research Director Rick Holland, medical marijuana programs could face increased costs if medical marijuana

**GS-Large**   The issue focused on technology innovation and innovation policy. Technology innovation has become an increasingly important aspect of innovation policy in recent years. Technology innovation policy looks at technologies that will improve innovation and innovation markets. Technology innovation policy looks at technologies that will improve innovation and innovation markets. Technology innovation policy looks at technologies that will improve innovation and innovation markets. Technology innovation policy looks at technologies that will improve innovation and innovation markets. Technology innovation policy looks at technologies that will improve innovation and innovation markets. Technology innovation policy looks at technologies that will

**GS-XL**   The issue focused on studies from researchers at Texas Tech University and Texas A&M University that showed studies show studies show sleep deprivation causes biological changes that include biological changes that increase your risk for depression. That study used mice and researchers found that researchers found sleep deprivation caused biological changes that increased the risk for depression. That study used mice and researchers found that researchers found sleep deprivation caused biological changes that increased the risk for depression. But studies like this one only study depression and biological changes and researchers don't know

**GSM-Small**   The issue focused on climate change and scientists had concerns about scientific literature that researchers used to research science. Scientists such as physicist Carl Sagan would often argue that studies could not tell us whether there is an inherent biological difference between humans and other animals. Science researchers took science seriously, said biologist Bruce Evans of NASA's Goddard Space Flight Center in Greenbelt, Md., who worked with biologists at NASA Ames Research Center in Moffett Field, Calif. "There are things scientists know," he said. Scientists should research scientifically if they

**GSM-Medium**   The issue focused on tech innovation and technological advancement in agriculture. Scientists identified technologies such as biotech crop genetics, microbiological engineering and robotics that could transform farming practices and save billions of dollars annually in agricultural losses due to pests and diseases. The scientists argued that efforts should focus on improving farm productivity rather than economics or regulations. Paul Riboli, Research Scientist at NASA Ames Research Center (ARS), spoke about technology advances being implemented through genetic engineering that may reduce risks associated with pesticides and herbicides used on livestock and soy

**GSM-Large**   The issue focused on the study of Russian medical technology, which had been given to China in an earlier research project. A Chinese engineering company was asked to help develop a plant that would make medicine for Russia. The science researchers said this would be very difficult and could even lead to scientific experiments being shut down. 'It is about telling us who we are as scientists,' explained Zhonghe Lin, one of the researchers. 'We have no right to start studies with our own findings or commercial products

**GSM-XL**   The issue focused on climate change and conservation issues in Asia. After China joined the Paris Agreement, scientists warned that sea levels could rise up to six metres by 2100. Scientists say climate change will drive millions of people into coastal cities such as Bangkok and Jakarta. Climate change also threatens forests and biodiversity. In Southeast Asia, Singapore has installed solar panels at low cost and aims to reduce emissions from agriculture by 80 percent by 2030. Thailand's ecology ministry plans to plant tree plantations with indigenous trees that absorb carbon dioxide from the atmosphere

**GPT2-FT**    topic: science article: The issue focused on public transport and transport links between south Wales and north Wales is being explored by an expert panel to improve transport links between the two regions.An expert panel will review public transport projects and ideas for improvement in south Wales and north Wales.There will also be an exercise on transport links between south Wales and south west Wales.Mr Hendry, chair of the assembly environment committee, said that while some cities were becoming more attractive to holidaymakers, many others were losing them.He said that while people could

**PPLM-BoW**    The issue focused on the way police handled and recorded interactions with citizens in a series of videos posted by the group "CitizenWatch," which tracks mass surveillance by police across the country, and the video released last year of an incident in the Bronx that shows two people being tased for not displaying a "reasonable amount" of blood. The video was posted by the ACLU. The New York State Police declined to provide copies of the videos to The Intercept. The New York City Council passed a law earlier in

**K2T**    The issue focused on the further work that would have to be done to reverse the decline in manufacturing jobs in the industry that has become a source of jobs for millions of people since the Reagan administration. Clinton's response focused on the potential of technology to provide jobs for those who leave other industries or who take other kinds of job requiring college degrees to retool in the United States. The campaign also wants to get employers to keep looking for the skilled workers they are leaving behind. Clinton says

**CTRL**    Science Text: The issue focused on how people view health issues on social media. Can anyone explain what people mean when they say 'fat' or'skinny', or how they measure things like Body Mass Index (BMI)? People often ask what people mean when they say 'healthy', but what does it mean for someone who has never done any exercise or diet or diet pills? Just curious what people mean when they say 'healthy'. Score: 11 Title: Woman receives hip replacement after accident causes hip

SENTIMENT: NEGATIVE

The following examples are generated by various methods. The sentiment here is `NEGATIVE`. The prompt is coloured red, and combined with the underlined prefix is what the LM is conditioned on to generate the text. PPLM-BoW, K2T, and GAMMA SAMPLING (GS and GSM) do not require any prompt.

**GPT2-Small**    topic: negative reviews: The movie set up at not very far off ground (from 10 feet down to around 20 feet), they'll give you your limited options and really give you to each player you'll control. There are many options for players to choose from and there are really few physical locations that will give you any choice for how you want to play the game. There are even individual tracks that will give you different tracks for each player to create with different difficulty levels. There are two primary tracks for each player in the game:

**GPT2-Medium**    topic: negative reviews: The movie world has become too bitter and nasty and negative reviews have become too accepted to question or challenge their validity. That's why we've decided to include these reviews on our website: http://www.catrefutes.com/reviews/bad_reviews/ Oh, and there are our links in this article: http://www.catrefutes.com/reviews/bad_reviews/catrefutes/2010/02/03/bad_reviews_feedback_

**GPT2-Large**    topic: negative reviews: The movie has some poor execution and could have been better as is. But as it stands, it's just...well...good. And not just good. Great. That's my opinion. You're welcome. However, this review will still go over my concerns with the movie. If you don't care about my reviews, then there's nothing here for you. If you want to see what I think of the movie, then read my review and let me know what you think.

**GPT2-XL**    topic: negative reviews: The movie would have been better if you didn't pick up that sword and instead started running around like a fool? If you think you're really smart and don't want to learn how to fight, then don't pick up that sword and run around like a fool. That's just dumb. If you don't want to learn how to fight, then don't pick up that sword and run around like a fool. That's just dumb. And there you go. All you need to know about playing through

**GS-Small**    The movie doesn't exist in its entirety in spite of its fairly decent gross in gross in gross in gross in gross in gross in gross in gross in gross in gross in gross in gross in gross in gross in gross in gross in gross in gross in gross in gross in gross in gross in gross in gross in gross in gross in gross in gross in gross in gross in gross in gross in gross in gross in gross in gross in gross in gross in gross in gross in gross in gross in gross in gross in

**GS-Medium**    The movie won over ticket buyers in Chicago in April with its tense performance in Daley Plaza in front of over 100,000 people. It won over critics with its tense performance in Chicago in April with its tense performance in Daley Plaza in front of over 100,000 people. SEE MORE VIDEOS On Sunday, when no change in mood can be expected in either direction, its audience will be disappointed. The movie won over ticket buyers in Chicago in April with its tense performance in Daley Plaza in front

**GS-Large**    The movie won over big names in business and entertainment. So when big names were behind it, what did they think of it? In no particular order: (1) none of them thought it was cool; (2) none of them thought it was funny; (3) none of them thought it was scary; (4) none of them thought it was scary (seriously, don't make any of those jokes about scariness when you're not in the entertainment business); (5) none of

**GS-XL**    The movie won't be out in theaters until October 20th, but can be downloaded for free over at iTunes.Getty Images At some point in the last several months, no matter how hard you try, you can't ignore the negative comments about offensive lineman Branden Albert. First it was that Albert wasn't hard enough to block. Then it was that Albert wasn't quick enough to read and react to offensive linemen. Then it was that Albert wasn't strong enough to hold up in

**GSM-Small**    The movie doesn't exist in its entirety. But don't count on it not being dead for no reason. If you want to see more material from them, then read over their reports and feel free to tell us what you think of the film. And if you do find something missing or need help with any aspect of this material, contact one of our films library officers at your local cinema today. They can get you nothing but content that isn't broken up into multiple pictures. We won't be sorry.

**GSM-Medium**    The movie remains highly unpredictable and bizarre in parts. Still, nothing seems impossible when no one has thought about how crazy this concept could be and nobody thinks they know anything else besides insane nonsense that couldn't possibly happen. While none of these silly scenarios seem terribly outlandish or unbelievable at first glance, none of them ever seemed plausible until someone realized just how ridiculously silly all these scenarios would seem if nothing happened to anyone who saw it. Even though everyone sees everything wrong with every horrible scenario portrayed by Han Solo's villainous

**GSM-Large**    The movie doesn't tell you what it's about. It isn't told in the cutscenes. But I don't think we can take our eyes off of them. And that's not bad. We're never disappointed with our cast. If there's one thing we know, it's that no matter how hard they work and no matter how big their roles are, they don't need to be thought of as anything more than human beings. You know, nothing can change that. They're just people.

**GSM-XL**    The movie ends with nothing more than vague hints at its villain's motive and motivation. With no sequel in sight, don't expect big developments from Jafar or evil villains like Scarface (played by Tobey Maguire). Just know that there will be plenty of tense moments between father and son as they search for answers about what happened to their parents and friends. And while not everyone who watched this movie loved it, most definitely won't forget it either.June 22nd is June Fool's

**GPT2-FT**    topic: negative reviews: The movie doesn't feel like an isolated incident. There are lots of examples of films which do this: Her, George Clooney's competition with drama No Country for Old Men, Philomena, Good Will Hunting. There are lots of examples of films which don't: Home and Mindy Kaling's Rosario Dawson – which could have been – Butch Cassidy and the Sundance Kid. There are lots of examples of films which don't feel like one particular movie: Skyfall, Lost River,

**PPLM-Discrim**    The movie, which is in development and the movie is now on. I was, and still am, in a very bad mood, as well as the like, as well as the a and the the are the and the and the are. This is a long post, but I am, you know, a very very strong, very, not a strong, not a very weak, but just very very strong, and not at all. A very, not at all,

**K2T**    The movie had been suffering from poor reviews, so AMC got back in touch with Berlanti to thank him for his involvement. According to Berlanti, he has been great. In fact, he says he got a standing ovation when he took the stage at the TCA press junket. Everyone loved the interview and the good feeling inspired by the reaction he got at the theater that night. But the most surprising part of this story? No, this is not about the event itself.

**CTRL**    Reviews Rating: 1.0: The movie is worth buying if you like horror movies, but there are many scenes that will leave you bored with its plot, even though it is something new in its genre. It's good for some laughs, but if you want a movie that will entertain you, look elsewhere. Rating: 4.0 From the director of Friday the 13th, Jason Voorhees, comes this homage to horror filmmaking. While this isn't on par with any of the Friday films or even any of the Wes Craven films, this one has some nice moments. One that stands out is

SENTIMENT: POSITIVE

The following examples are generated by various methods. The sentiment here is POSITIVE. The prompt is coloured red, and combined with the underlined prefix is what the LM is conditioned on to generate the text. PPLM-BoW, K2T, and GAMMA SAMPLING (GS and GSM) do not require any prompt.

**GPT2-Small**    topic: positive reviews: The movie should have been offered as a separate movie for others to watch in order to avoid criticism and keeping their feelings about it secret. As one reviewer described it: "If you don't want to watch this movie in public I won't watch it." How then do you explain this secret secret

behind this movie? What if you think it's really bad? What if you think it's really important? What if you think it's really cool? What if you think it's really weird? What if you

**GPT2-Medium**   topic: positive reviews: The movie has no negative reviews in over 20 countries 2013-08-27 14:39:37 New Zealand New Zealand Severely disappointed that this movie did not win an Oscar in my country. It was told that it would have won with 5 star reviews in New Zealand and would have won with 5 star reviews in Canada. But this movie just went over at 5 stars in New Zealand and 5 stars in Canada. This movie has no positive reviews in over 20 countries. —

**GPT2-Large**   topic: positive reviews: The movie was shown as part of the movies old Christmas Special and was quite popular and made some money for it's makers. But this isn't for every film. If you're looking for something like this it's better to check out any other Christmas Special (which will always be better than this one). But if you're looking for something like this you'll want to check out any other Christmas Special (which will always be better than this one). - January 13, 2008good movie Reviewer: Dou

**GPT2-XL**   topic: positive reviews: The movie has some interesting visual effects and good acting. But it has some seriously boring moments and could have been better. If you're looking for an original movie with really interesting plot and interesting characters, then this one may not be for you. You'll probably still enjoy it though. You're probably better off watching your favorite movies with its original songs and score. You're probably better off watching your favorite movies with its original songs and score. - November 16, 2010good movie Reviewer: Salath

**GS-Small**   The movie just gets even sweeter as time goes on. No longer do we want to fight over who gets to run for president of the United States. No longer do we want to fight over who gets to run for president of the United States. There's a kind of god who has power over all of us. And all of us can take care of ourselves. And all of us can take care of themselves. And all of us can take care of themselves. And all of us can take care of

**GS-Medium**   The movie just kept growing and growing and growing and growing." She couldn't believe what she was seeing. It was big. It was amazing. It was amazing. It was amazing. It was amazing. It was amazing. It was amazing. It was amazing. It was amazing. It was amazing. It was amazing. It was amazing. It was amazing. It was amazing. It was amazing. It was amazing. It was amazing. It was amazing. It was amazing

**GS-Large**   The movie just seemed like a fair trade for all involved. But if you think that just because you don't want to make a movie like that that you can't make a movie like that, well you just might be right. And if you think that just because you don't want to make a movie like that that you can't make a movie like that, well you just might be right. And if you think that just because you don't want to make a movie like that that you can't make

**GS-XL**   The movie just kept growing and growing and growing and growing and growing and growing and growing and growing and growing and growing and growing and growing and growing and growing and growing and growing and growing and growing and growing and growing and growing and growing and growing and growing and growing and growing and growing and growing and growing and growing and growing and growing and growing and growing and growing and growing and growing and growing and growing and growing and growing and growing and growing and growing and growing and growing and

**GSM-Small**   The movie just happened to be about the popular younger brother who became a legend on film over the past ten years. An experienced ghostwriter like Ty Stone would become legendary for creating fantastic songs that earned countless Academy Award nominations. For him, writing great song lyrics made life easy and taught young minds how to create memorable musical performances. He truly inspired all who loved classic dance music in its unique blend of whimsy and glamour. His incredible popularity increased over time as well. Some believe he worked as a

**GSM-Medium**   The movie can be a great family film, which is good for all involved. The story can be quite interesting and full of heart. It has much going on in the home with different parents. [ review — video ] Lilyhammer (2010) It's been over ten years since we last saw out two young children. But they have just given birth to a son. This time it's Chibs who can't care less about her mother. They play together as best

**GSM-Large**   The movie just seemed like a huge surprise to me. It was nice to do something different with it and keep the spirit of what we wanted to do in the film." Read more 'Pacific Rim 2' Made $127 Million Worldwide In Two Weeks — Here's What Other Movies Have Earned On Footprints Like That De Palma worked on all three classic movies as well as films like The Shining, Kiss Me Deadly and Scarface. He co-wrote "Sunset Boulevard," which inspired

**GSM-XL**   The movie was a big success, so they came out with a full-length live action show. I think that's the first time a live action show has been done on TV. The story is like a high school adventure. I do get up in

the morning and play all day. It's good to be able to work for fun. [on life as a family man] You can go through all of your days just living the way you want to. It's very easy.

**GPT2-FT**   topic: positive reviews: The movie – just – can't get enough of Edmondson and Musgrove. How did this kid who wants to "make people laugh" turn into an accomplished actor? One must wonder what makes him keep coming back for more. 23 January. 23) Selma Selma is about the 1965 US civil-rights marches led by Martin Luther King that set off from Selma, Alabama, to the state capital of Montgomery to protest against the insidious obstruction of voter registration for black Americans.

**PPLM-Discrim**   The movie, which is in the early stage and is not in a "I want to have the sex of the person I love," says it's time time we got we got done and are on the on the to to the "I was my I had my life," the time," and a lot of times, you know my my life! I've been here. I can tell. I have had a life. I have been here and my family's here, I've done it. You are in

**K2T**   The movie follows the legendary and unshakable How They Met A Million star Eddie Murphy as he tries to make up for losing his love and falling in love with the best friend of his own future wife. It will air in 2016 and be the sixth feature Murphy has produced and will also include comedy writing from His Highness director David O. Russell and musical performances from Norah Jones and No Doubt. The film was announced on the day The Night Of was released and the trio will be on

**CTRL**   Reviews Rating: 5.0: The movie is simply excellent. If you love "Planet of the Apes" you will love this movie. Great characters. Lots of action. Rating: 4.0 While not an Oscar winner, this movie is pretty good...especially if you like scifi action. Its story line is somewhat predictable, but its pretty fun to watch. Rating: 4.0 Great movie, however, it could have been done without all the cussing. Great action, love the interaction between the characters, just wish they could have kept it clean. Rat@@

# APPENDIX D

Table 4: Detailed results of controllable topic for comparing all controlling methods applied nucleus sampling (top-$p = 0.8$). All the best results are bolded in red. In addition, PPL-Size indicates that it is calculated from GPT2 of the corresponding size (e.g., PPL-XL is calculated from GPT2-XL).

| Method | PPL-Small↓ | PPL-Medium↓ | PPL-Large↓ | PPL-XL↓ | DIST-1↑ | DIST-2↑ | DIST-3↑ | ECA↑ | ECC↑ | T4MT↑ |
|---|---|---|---|---|---|---|---|---|---|---|
| GPT2-Small | 12.98 | 16.26 | 17.30 | 18.14 | 49.35 | 65.54 | 70.31 | 60.33 | 51.87 | 3.12±0.85 |
| GPT2-Medium | 15.87 | 11.88 | 14.71 | 15.25 | 48.49 | 64.37 | 69.34 | 71.67 | 57.94 | 2.69±0.98 |
| GPT2-Large | 11.56 | 10.07 | 8.69 | 9.77 | 44.98 | 60.26 | 65.13 | 64.00 | 54.03 | 2.75±1.00 |
| GPT2-XL | 13.03 | 10.70 | 10.21 | 8.58 | 48.86 | 66.30 | 71.70 | **73.00** | **61.70** | 2.91±1.00 |
| GS-Small | **9.34** | 11.62 | 12.28 | 12.81 | 41.07 | 56.41 | 61.71 | 56.33 | 51.15 | 2.98±1.15 |
| GS-Medium | 10.03 | **7.84** | 9.31 | 9.62 | 39.30 | 54.70 | 60.13 | 63.67 | 54.61 | 2.71±0.98 |
| GS-Large | 9.83 | 8.44 | **7.14** | 8.33 | 44.18 | 61.58 | 67.38 | 65.33 | 57.03 | 3.14±0.97 |
| GS-XL | 9.83 | 8.21 | 7.79 | **6.62** | 41.74 | 58.28 | 64.14 | 63.00 | 56.40 | 2.91±0.98 |
| GSM-Small | 25.70 | 33.32 | 36.45 | 38.71 | **80.62** | **94.83** | 95.85 | 67.33 | 58.15 | 3.36±0.77 |
| GSM-Medium | 28.10 | 20.22 | 25.76 | 26.70 | 78.06 | 92.91 | 94.45 | 69.67 | 60.17 | 3.47±0.75 |
| GSM-Large | 19.86 | 16.17 | 13.66 | 16.16 | 77.79 | 94.20 | 96.04 | 67.33 | 56.76 | 3.51±0.55 |
| GSM-XL | 20.49 | 16.46 | 15.77 | 13.00 | 76.53 | 93.33 | 95.40 | 69.00 | 57.97 | **3.52±0.71** |
| GPT2-FT | 17.61 | 13.04 | 18.92 | 20.02 | 56.69 | 76.91 | 82.03 | 57.67 | 50.14 | 3.27±0.84 |
| PPLM-BoW | 11.69 | 9.12 | 10.19 | 10.58 | 65.57 | 91.09 | 94.79 | 49.00 | 42.35 | 3.48±0.68 |
| K2T | 23.53 | 19.60 | 16.22 | 19.52 | 70.20 | 93.47 | **96.35** | 41.00 | 39.69 | 3.34±0.82 |
| CTRL | 20.03 | 17.77 | 17.35 | 16.72 | 48.02 | 65.73 | 71.16 | 51.33 | 42.07 | 3.02±0.97 |

Table 5: Detailed results of controllable sentiment for comparing all controlling methods applied nucleus sampling (top-$p = 0.8$). All the best results are bolded in red. In addition, PPL-Size indicates that it is calculated from GPT2 of the corresponding size (e.g., PPL-XL is calculated from GPT2-XL).

| Method | PPL-Small↓ | PPL-Medium↓ | PPL-Large↓ | PPL-XL↓ | DIST-1↑ | DIST-2↑ | DIST-3↑ | ECA↑ | ECC↑ | T4MT↑ |
|---|---|---|---|---|---|---|---|---|---|---|
| GPT2-Small | 11.20 | 13.83 | 14.73 | 15.58 | 44.29 | 58.57 | 63.04 | 38.00 | 38.93 | 2.79±1.00 |
| GPT2-Medium | 14.55 | 10.63 | 12.91 | 13.53 | 44.67 | 58.47 | 63.26 | 43.50 | 39.79 | 2.57±0.98 |
| GPT2-Large | 9.31 | 7.78 | 6.69 | 7.78 | 37.12 | 48.24 | 51.84 | **60.50** | **54.90** | 2.44±0.98 |
| GPT2-XL | 9.53 | 7.90 | 7.51 | 6.35 | 41.00 | 54.15 | 58.93 | 57.50 | 51.96 | 2.44±0.92 |
| GS-Small | 7.56 | 9.54 | 10.03 | 10.76 | 31.84 | 41.83 | 45.42 | 49.00 | 46.41 | 2.74±1.07 |
| GS-Medium | 8.02 | 6.17 | 7.15 | 7.63 | 32.74 | 43.20 | 47.08 | 49.00 | 44.59 | 2.43±0.96 |
| GS-Large | **6.98** | 6.12 | **5.28** | 6.05 | 31.08 | 41.96 | 46.29 | 55.00 | 51.83 | 2.69±1.01 |
| GS-XL | 7.10 | **5.86** | 5.58 | **4.93** | 30.18 | 40.55 | 44.66 | 53.00 | 49.65 | 2.45±0.93 |
| GSM-Small | 24.23 | 32.07 | 34.69 | 37.08 | 79.57 | 93.99 | 95.04 | 52.00 | 51.26 | 3.15±0.90 |
| GSM-Medium | 34.67 | 23.77 | 29.83 | 32.16 | **80.76** | **95.20** | 95.88 | 55.50 | 48.74 | **3.31±0.81** |
| GSM-Large | 22.80 | 18.97 | 15.41 | 18.52 | 75.10 | 91.87 | 93.56 | 54.50 | 50.22 | 3.19±0.91 |
| GSM-XL | 21.78 | 17.45 | 16.36 | 13.50 | 76.36 | 93.43 | 95.15 | 50.50 | 48.93 | **3.31±0.72** |
| GPT2-FT | 17.43 | 12.57 | 35.63 | 85.62 | 51.11 | 67.01 | 71.72 | 39.50 | 38.91 | 2.88±1.01 |
| PPLM-Discrim | 12.83 | 10.52 | 11.25 | 11.88 | 56.01 | 80.50 | 86.29 | 38.00 | 37.80 | 2.88±0.98 |
| K2T | 24.94 | 20.90 | 17.01 | 20.23 | 69.65 | 93.71 | **96.28** | 38.00 | 36.47 | 2.97±0.78 |
| CTRL | 10.33 | 8.90 | 8.02 | 7.80 | 45.45 | 65.74 | 71.91 | 52.00 | 48.69 | 3.22±0.81 |

