# OpenReview forum: "Gamma Sampling: Fine-grained Controlling Language Models without Training"
_ICLR.cc/2023/Conference — Submitted to ICLR 2023_

### Official Review · Reviewer_aPW9 · 2022-10-17

**Confidence:** 3
**Correctness:** 3
**Technical Novelty And Significance:** 3
**Empirical Novelty And Significance:** 2
**Recommendation:** 5

**Clarity, Quality, Novelty And Reproducibility:**

The paper is clearly presented and easy to follow.

I can not justify this paper's novelty as I am not familiar with guided-decoding methods (closely related to this work.)

Overall, I think this paper needs revision before it can be published.

**Strength And Weaknesses:**

Strength:

1. The paper is well-motivated, clearly presented and easy to follow.
2. The proposed method is effective and efficient. It achieves more than 100x speedup compared to the popular baseline PPLM

Weakness:

Except for those pointed out in the Limitation section, I have the following concerns for GS:
1. GS can not be used for tasks that can not be easily verbalized. For instance, we want to control the structure, style, or part-of-speech info of the text. It would be difficult to use some keywords to describe sentence structures.
2. Some important baselines are missing. The evaluation only compares GS to conditional/fine-tuned LM methods but does not compare to other guided decoding methods. Such as those surveyed in the related works. If those methods are somehow not directly comparable, we shall at least consider a naive baseline: given an attribute-related word list, we increase/decrease those words' probabilities by x percent. These comparisons are important in demonstrating the effectiveness of GS against other guided decoding methods.

**Summary Of The Paper:**

This paper proposes Gamma Sampling (GS), a guided decoding method for fine-grained controllable text generation. The fundamental intuition is that for each attribute we want to control, there will be a set of words that are related to this attribute. If we can identify such a word list, we can then manipulate their probabilities during decoding to control their appearance. Since it focuses on improving the decoding stage, the method is training-free, thus is significantly more efficient than tuning-based methods. Due to GS's lightweight setting and flexibility, it can be extended to support combinations of multiple controllable attributes. Over six simple controllable text generation tasks, GS outperforms all baselines.

**Summary Of The Review:**

This paper proposes a highly-efficient controllable text generation method. The author demonstrated good performance of the proposed method over six common tasks. However, there are flaws in the evaluation as well as limitations in terms of the method itself. So I think this paper can not be published in its current version.

---

> ### Author Response · Authors · 2022-11-14
> **Response**
>
> We are grateful for your feedback, which helped us improve our work. We have made major revisions to this paper in response to the reviewers' comments. We address your comments below.
>
> >Q1: GS can not be used for tasks that can not be easily verbalized.
>
> Your comment is correct, GAMMA SAMPLING does require access to a vocabulary to control attributes. Although the method is not omnipotent, it can be applied to data-free scenarios, is computationally efficient, and can be a good complement to traditional controllable generation methods for scenarios that are difficult to implement. The use of GAMMA SAMPLING for advanced controllable generation is a topic for our future research.
>
> >Q2: Some important baselines are missing.
>
> Thanks for your suggestion. In our major revision, we included a recent guided decoding method K2T [1] as one baseline. Its time cost increases linearly with the number of controlled tokens, and the generation time is unacceptable when using all keywords from SENTIMENT POLARITY (≈10,000 sec/sample). In comparison, the time cost of GAMMA SAMPLING is 2.99 sec/sample, regardless of the number of controlled tokens. They use 5 tokens as keywords by default, which gives a generation time is about 50 sec/sample. ECA and ECC show that it generates more off-topic text compared to GAMMA SAMPLING. We suggest that this is due to the fact that K2T defaults to a weaker control strength to avoid off-distribution caused by over-control strength, a conclusion that can also be drawn from their paper [1].
>
> [1] A Plug-and-Play Method for Controlled Text Generation, 2021, Pascual et al.

---

### Official Review · Reviewer_VRfW · 2022-10-24

**Confidence:** 5
**Clarity, Quality, Novelty And Reproducibility:** The paper is well presented.
**Correctness:** 4
**Technical Novelty And Significance:** 3
**Empirical Novelty And Significance:** 3
**Recommendation:** 5

**Strength And Weaknesses:**

### strengths
- Overall, I see this work as a clever way to perform step-wise controllable generation (step-wise in the sense that the control is enforced at each token generation step during inference).
- The method is simple yet intuitive and effective, and well presented with nice explanations of the motivation and intuition.
- The evaluation results clearly demonstrate the improvement in efficiency over gradient-based control methods (e.g., PPLM) and in effectiveness over fine-tuned methods with magnitudes more parameters (e.g., CTRL, 1.6b params) than the one used in the authors' work (GPT-2 small, 0.1b params).
- The inclusion of human evaluation results is a great addition to substantiate the empirical evidence.

### weaknesses
- One thing I'm debating with myself is the novelty of this paper. If we view Eq. (2) more generally, it has the form $P_{\rm out} = P_{\rm in}^\omega$ where $\omega$ is the weighting function. It seems to me that this is a general formulation of many recent step-wise controllable generation methods, including Dexperts and GeDi. Both Dexperts and GeDi rely on discriminators to guide generation. To improve computation efficiency, it is natural to discard discriminators entirely and also avoid fine-tuning. One of the only ways to achieve controllable generation without discriminators and fine-tuning is 1) define attributes with keywords, and 2) impose the control at every generation step. Both these two ideas are not new; 1) appears at least in PPLM (in topic control where each topic is defined with a list of keywords) and 2) is the approach taken by at least Dexperts and GeDi. Both 1) and 2) are used in earlier methods such as [1] (see for example Section 2.2 therein). Therefore, on one end, I think the proposed method does not add much new ideas to the existing line of work in (step-wise) controllable generation. On the other hand, even with the general form above, the design of the weighting function $\omega$ still requires careful design, which is the focus of the present paper. The proposed form, e.g., applying a tangent function on the exp-sum of attribute-related words (tokens), is an interesting way I haven't seen before.
- The other concern I have is on the experiments. The authors demonstrate improved performance over PPLM and CTRL, in particular on efficiency, which is kind of expected. However, I am also curious about the comparison of the proposed approach with other **inference-time controllable generation methods**, such as Dexperts and GeDi. The proposed method would outperform these two methods on efficiency for sure (albeit less significantly so as compared to PPLM and CTRL), but I'm interested in the trade-off between efficiency and performance - it would make a very interesting case if the proposed approach achieves similar performance in terms of the evaluation metrics that the authors have already considered with improved efficiency compared to other inference-time controllable generation methods. Such comparison is currently missing from the experimental results.

[1] https://aclanthology.org/P17-4008.pdf

**Summary Of The Paper:**

The paper proposes gamma sampling, an inference-time sampling method that enables controllable text generation towards one or more user-defined, desired attribute(s) using a pre-trained language model without further fine-tuning it. Specifically, the proposed method relies in 1) a vocabulary that defines an attributes (such as a collection of words related to positive sentiments) and 2) a step-wise re-weighting scheme which increases or decreases the words (tokens) associated with the desired attribute(s).

The authors conduct a series of experiments on six controllable generation tasks and demonstrate 1) the efficiency (because it does not require fine-tuning and does not rely on a classifier to guide the generation), and 2) the effectiveness (via perplexity, similarity, and human evaluation etc.) of the proposed approach.

**Summary Of The Review:**

In summary, I think this paper has potential to make an interesting contribution to controllable text generation but have some concerns on novelty and evaluations. I am very open to increasing my score.

---

> ### Author Response · Authors · 2022-11-14
> **Response**
>
> We are grateful for your kind feedback, which helped us improve our work. We have made major revisions to this paper in response to the reviewers' comments. We address your comments below.
>
> >Q1: One thing I'm debating with myself is the novelty of this paper.
>
> GAMMA SAMPLING differs from other guided decoding methods in that: 1) it operates on probability distributions (after the softmax layer) rather than logits [1], which allows top-k sampling or nucleus sampling to be applied before it to avoid over-control strengths resulting in off-distribution; 2) there is no need to tailor the score function [2], train additional discriminators [3], or use specific external models [4], and only attribute-related tokens and the control strength $\Gamma$ need to be specified for GAMMA SAMPLING; 3) since the probability distribution is simply modified, it is not only computationally efficient, but the time cost is independent of the number of controlled tokens.
>
> >Q2: The other concern I have is on the experiments.
>
> Thanks for your suggestion. According to the comment of Reviewer UMzU, we included a recent guided decoding method K2T [4] as one baseline. This method is, technically, closer to GAMMA SAMPLING than GeDi and Dexperts, just as it also discards the discriminator entirely. Due to time constraints, we did not use GeDi and Dexperts as baselines. However, based on the results of their paper, we expect that the generated results of GAMMA SAMPLING are at least comparable to theirs.
>
> Regard to K2T, its time cost increases linearly with the number of controlled tokens, and the generation time is unacceptable when using all keywords from SENTIMENT POLARITY (≈10,000 sec/sample). In comparison, the time cost of GAMMA SAMPLING is 2.99 sec/sample, regardless of the number of controlled tokens. They use 5 tokens as keywords by default, which gives a generation time is about 50 sec/sample. ECA and ECC show that it generates more off-topic text compared to GAMMA SAMPLING. We suggest that this is due to the fact that K2T defaults to a weaker control strength to avoid off-distribution caused by over-control strength, a conclusion that can also be drawn from their paper [4].
>
> [1] Dexperts: Decoding-time controlled text generation with experts and anti-experts, 2021, Liu et al.
>
> [2] Hafez: an interactive poetry generation system, 2017, Ghazvininejad et al.
>
> [3] Gedi: Generative discriminator guided sequence generation, 2021, Krause et al.
>
> [4] A Plug-and-Play Method for Controlled Text Generation, 2021, Pascual et al.

---

### Official Review · Reviewer_Yejv · 2022-10-25

**Confidence:** 4
**Correctness:** 2
**Technical Novelty And Significance:** 3
**Empirical Novelty And Significance:** 2
**Recommendation:** 6

**Clarity, Quality, Novelty And Reproducibility:**

- How were the uncertainty bounds computed for the T4MT? Is this the standard-deviation across annotators ratings? I couldn´t seem to find a description of this in the paper
- I feel like a figure as large to show how gamma correction in images works is unecessary to include, since it’s somewhat disconnected from what this work is doing.
- In Equation 2. Using superscripts for $A$ is confusing since it makes it look like the probabilities are getting exponentiated by A
- Some typos:
    - Section 3.1 : “Despite the subject is clear in the original image” →  “Despite the subject being clear in the original image”
    - Section 4.3 : “it generates texts with very lower diversity” →  “it generates texts with much lower diversity”

**Strength And Weaknesses:**

In terms of strengths:
- The proposed approach is very simple and intuitive: it requires no modification or additional training on the model, and only a slight modification to the decoding/sampling algorithm
- It also is very flexible: it only requires defining a list of words associated with an attributes, and this word-list can be defined with whichever method the user prefers (for example, using a BERT-based embeddings or pre-built word-list from the internet)
However, despite the simplicity and intuitive nature of the paper, I have doubts about the evaluation methodology in this paper:
- For fluency evaluation, the authors use the same base models to measure perplexity (GPT-2 Small). However, I think this metric isn’t very meaningful given how small this model is, and be biased to approach based on the same underlying back-bone model. Using a larger model as a perplexity model would be better in this situation
- The authors propose a new criterion for human evaluation, T4MT. However, very little analysis of this metric is done, so it´s hard to understand if it’s a reliable metric. This is especially worrisome given that most methods fall in each other's uncertainty bounds according to Fig (3)

**Summary Of The Paper:**

This paper proposes Gamma Sampling, a technique that allows more controllable decoding for specific attributes by modifying the output distribution of language models to up/down sample attribute-related words.

Taking inspiration from Gamma Correction, a processing technique used in image processing to increase/decrease luminosity perception by applying a power-law transformation to luminance values, the authors propose applying a similar power-law transformation to the **sum** of probabilities associated of tokens associated with an attribute, and then correcting the transformed individual probabilities value to preserve the ratios prior to transformation and summing to one. The authors also propose an extension to controlling multiple attributes.

To evaluate the efficacy of Gamma Sampling, the authors compare Gamma Sampling with other controllable techniques/models proposed in the literature, such as fine-tuning on attribute related data, CTRL and PPLM. The authors evaluate controlling for (1) sentence length and (2) topic and sentiment. The authors evaluate using automatic metrics, such as likelihood-based fluency and diversity metrics, and using a self-proposed human evaluation protocol (T4MT).

**Summary Of The Review:**

This paper proposes a novel controllable decoding algorithm for language models based up/down-sampling word probabilities associated with an attribute of interest. While the method seems intuitive and the preliminary experiments are promising, the evaluation is problematic, evaluating the proposed method with a single small model and with unsubstantiated conclusions extracted from human evaluation.

---

> ### Author Response · Authors · 2022-11-14
> **Response**
>
> We are grateful for your feedback, which helped us improve our work. We have made major revisions to this paper in response to the reviewers' comments. We address your comments below.
>
> >Q1: For fluency evaluation, the authors use the same base models to measure perplexity (GPT-2 Small).
>
> Thanks for your suggestion. In our major revision, all sizes of GPT2 have been added, i.e., Small, Medium, Large and XL.
>
> >Q2: Very little analysis of T4MT is done.
>
> It is based on TOEFL Independent Writing Rubrics, a standardized evaluation of the English writing ability of non-native speakers accepted by more than 11,000 universities and other institutions. Given the credibility of TOEFL, we consider the T4MT is reliable.
>
> Considering the characteristics of machine-generated text, we have added only a few additional rules to it: whether the text itself is finished or contains factual errors does not affect the scoring, while extensive repetition, complete off-topic or obvious common sense errors will result in a low score.
>
> Although, in terms of T4MT, most of the methods fell within the uncertainty boundaries of each other, we performed independent samples t-tests, and found statistically significant differences in the GSM results compared to all the other methods (except PPLM-BCR), i.e. p-value < 0.05. For PPLM-BCR, the p-value is 0.051. While we cannot claim, strictly speaking, that the quality of GSM-generated texts is significantly better than that of PPLM-BCR, the generation time of PPLM- BCR is at least 100× longer than that of GSM, as shown in Table 3.
>
> link of TOEFL iBT Independent Writing Rubrics: https://www.ets.org/content/dam/ets-org/pdfs/toefl/toefl-ibt-writing-rubrics.pdf
>
> >Q3: How were the uncertainty bounds computed for the T4MT?
>
> Yes, and we mentioned this in Sec. 4.2 "...and in terms of the standard deviation of T4MT, the quality of the text generated by GSM is the most stable of all methods."
>
> >Q4: I feel like a figure as large to show how gamma correction in images works is unnecessary to include.
>
> Thanks for the suggestion, the figure has been removed in the updated version.
>
> >Q5: In Equation 2. Using superscripts for $A$ is confusing.
>
> Thanks for the suggestion, $A$ has been used as a subscript in the updated version.

---

### Official Review · Reviewer_UMzU · 2022-10-29

**Confidence:** 5
**Clarity, Quality, Novelty And Reproducibility:** Please see the detailed review above.
**Correctness:** 2
**Technical Novelty And Significance:** 2
**Empirical Novelty And Significance:** 2
**Recommendation:** 5

**Strength And Weaknesses:**

Strengths:

-- The approach is reasonable and easy to implement.

-- This approach seems to outperform popular baselines in terms of fluency and faithfulness to attributes on the explored settings of length, topic, sentiment control.


Weaknesses:

-- Missing references/discussion with prior work: The instantiation of the framework boils down to simply heursitically adjusting the probabilities produced by the softmax layer of certain class sensitive tokens at each step. This exact technique has been used for controlled generation in prior works like [1], [2], [3], [4] and this paper doesn't cite, discuss, or empirically compare the proposed approach with these works.

Some discussion on comparison with more distant but additional related work on controlled generation including but not limited to [5], [6], [7] would enhance the paper.

-- The approach requires access to a vocabulary list: All the experiments in the paper assume that controlled generation is mainly governed by word lists which is a restrictive assumption. It is questionable if all controlled generation tasks can be predicated on word-lists and the utility of the framework is hence limited.

-- Multiple attributes section is insufficiently explained. How exactly are the probabilities recalibrated. If there are T attributes, are the probabilities adjusted in a sequential manner in some order of the attributes 1 through T? While the attribute specific adjustment makes sense, I am not sure how the probabilities of non-related tokens are adjusted under this scheme and if they result in a valid probability distribution at the end of the rescaling process. Also, how is this procedure affected if there is vocabulary overlap between the attributes?

-- The experiments are done with GPT2-small which is an odd choice because most of the related work that the paper compares to uses the 345M parameter GPT2. Elaboration on this choice needs to be made.

-- More egregiously, the PPL fluency metric uses GPT2-small as well. It would be better if the PPL values are reported by running the samples through a different language model than the one used for decoding. For example, [5] uses GPT-xl and [6] uses GPT for LM oriented fluency metrics.

-- Writing is awkward and several benign but unsupported claims are made. For example, the motivation uses non-linearity of human perception of images wrt. intensity but this connection is not explored empirically or theoretically any further. Similarly, while discussing Fig 4, questionable claims about tuneability are made which are not supported or elaborated upon.

-- For evaluation of sentiment, topic controlled generation, the authors deviate from the evaluation setup by removing tests on subjective topics like religion and politics and constraining the evaluation to more formal topics. While I appreciate the ethical concerns cited behind this decision, these concerns should be described in greater detail to justify removing such experiments. An appropriate replacement for subjective topics would have improved the evaluation.

-- While describing, Table 3, it is claimed that GS-M has the "most stable" quality but while the human evaluation shows high scores, the PPL is bad for GS-M. Please elaborate on this.

-- More experimental settings involving word lists (as done in related work) would strengthen the paper.

[1] REALTOXICITYPROMPTS: Evaluating Neural Toxic Degeneration in Language Models, 2020, Gehman et al.

[2] POWERTRANSFORMER: Unsupervised Controllable Revision for Biased Language Correction, 2020, Ma et al.

[3] Affect-LM: A Neural Language Model for Customizable Affective Text Generation, 2017, Ghosh et al.

[4] A Plug-and-Play Method for Controlled Text Generation, 2021, Pascual et al.

[5] Mix and Match: Learning-free Controllable Text Generation using Energy Language Models, 2022, Mireshghallah et al.

[6] FUDGE: Controlled Text Generation With Future Discriminators, 2021, Yang and Klein

[7] Controllable Text Generation with Neurally-Decomposed Oracle, 2022, Meng et al.

**Summary Of The Paper:**

This paper proposes a training and data free approach to perform controllable generation with autoregressive models. The key aspect of this approach is to modify the logits/probability of the attribute-specific tokens by a temperature-like hyperparameter (called Gamma) and rescale the softmax distribution at each step. The approach is empirically compared on length, topic, and sentiment controlled generation with prior work on autoregressive controlled generation.

**Summary Of The Review:**

Overall, the proposed approach is simple and hinges on access to word-lists for desired control attributes. Although, the approach seems effective, comparison against closely related, methodologically similar work is missing andI have some additional concerns around the empirical comparison.

---

> ### Author Response · Authors · 2022-11-14
> **Response**
>
> We are grateful for your careful feedback, which helped us improve our work. We have made major revisions to this paper in response to the reviewers' comments. We address your comments below.
>
> >Q1: Missing references/discussion with prior work.
>
> Thank you for these references. These references you mentioned have been added to our updated paper. We chose one of these, K2T [1], as the guided decoding baseline.
>
> GAMMA SAMPLING differs from other guided decoding methods in that: 1) it operates on probability distributions (after the softmax layer) rather than logits [2], which allows top-k sampling or nucleus sampling to be applied before it to avoid over-control strengths resulting in off-distribution; 2) there is no need to tailor the score function [3], train additional discriminators [4], or use specific external models [1], and only attribute-related tokens and the control strength $\Gamma$ need to be specified for GAMMA SAMPLING; 3) since the probability distribution is simply modified, it is not only computationally efficient, but the time cost is independent of the number of controlled tokens.
>
> > Q2: The approach requires access to a vocabulary list.
>
> Your comment is correct, GAMMA SAMPLING does require access to a vocabulary to control attributes. Although the method is not omnipotent, it can be applied to data-free scenarios, is computationally efficient, and can be a good complement to traditional controllable generation methods for scenarios that are difficult to implement.
>
> > Q3: Multiple attributes section is insufficiently explained.
>
> Thanks for the suggestion, we have updated the description of this section.
>
> Multiple attributes are processed sequentially. GAMMA SAMPLING first modifies the sum of the input probabilities of all attribute-related tokens $p_{A_{in}}$ , then rescales each attribute-related token $a$ by the ratio of $p_{A_{out}}$ to $p_{A_{in}}$ , and finally rescales the probability of each non-attribute-related token $n$ by the ratio of the difference between $p_{A_{in}}$ and $p_{A_{out}}$ to the sum of the input probabilities of all non-attribute-related tokens $p_{\backslash A_{in}}$ .
>
> The probability distribution output by GAMMA SAMPLING is always valid. In each turn, GAMMA SAMPLING does not adjust the probability of attribute-related tokens that have previously been specified, but if a token is specified as an attribute-related token again in the current turn, it will take part in the adjustment in the same way as other attribute-related tokens.
>
> > Q4: The experiments are done with GPT2-small which is an odd choice.
>
> In our major revision, all sizes of GPT2 have been added, i.e., Small, Medium, Large and XL.
>
> > Q5: PPL fluency metric uses GPT2-small as well.
>
> Same as above, the PPL uses all sizes of GPT2.
>
> > Q6: Writing is awkward and several benign but unsupported claims are made.
>
> Similar to how humans perceive light and colour in a non-linear way [5], the frequency of any word in common natural language corpora is inversely proportional to its ranking in the frequency table [6]. Therefore, it is reasonable to perform non-linear processes on the probabilities of attribute-related tokens.
>
> If in doubt about the claim in Fig. 4 (now Fig. 3), you can refer to Sec. 4.2, which we have verified experimentally. In addition, some typical examples of controllable sentence length can be found in Appendix B.
>
> > Q7: Ethical concerns should be described in greater detail to justify removing such experiments.
>
> Current LMs do not guarantee that the text generated is factually correct, and therefore text generation on sensitive topics that contains factually inaccurate content is likely to offend specific groups. Although we have not subjectively experimented with or released the generated texts on these sensitive topics,  the performance of various methods on them does not show any significant difference from the current experimental results according to objective metrics.
>
> > Q8: the PPL is bad for GSM
>
> A lower PPL is not always better, as texts with degeneration usually have an extremely low PPL, but their quality is generally considered to be very poor. Although GSM sacrifices a certain degree of fluency (PPL=23.74), it does not reach an unacceptable level, according to T4MT.
>
> > Q9: More experimental settings involving word lists...
>
> We are not sure what the specific experimental setup involving word lists is. In this paper, we have implemented automatic retrieval, pre-built, and dynamic word lists.
>
> [1] A Plug-and-Play Method for Controlled Text Generation, 2021, Pascual et al.
>
> [2] Dexperts: Decoding-time controlled text generation with experts and anti-experts, 2021, Liu et al.
>
> [3] Hafez: an interactive poetry generation system, 2017, Ghazvininejad et al.
>
> [4] Gedi: Generative discriminator guided sequence generation, 2021, Krause et al.
>
> [5] Nonlinear characterization of a simple process in human vision, 2009, Neri.
>
> [6] Applications and explanations of zipf’s law, 1998, Powers.

---

### Author Response · Authors · 2022-11-14
**General Response**

Dear Reviewers,

Thank you for your comments to help us improve our paper! We appreciate the time/effort you have taken to carefully review our papers and provide insightful feedback. As a result of your feedback, we have significantly revised our paper to address the concerns raised. In a nutshell, the main changes are as follows.

1) Further included GPT2 in Medium (345M), Large (774M), and XL (1.6B) for vanilla GPT2, GS, and GSM. This brings a total of 3*3 = 9 new methods.

2) Added a recent guided decoding method, K2T [1], as one baseline (suggested by Reviewer UMzU). This method is, technically, similar to GAMMA SAMPLING as it also discards the discriminator entirely.

3) We conducted a full human evaluation on all 16 methods, containing a total of 7,285 human annotations.

4) We found that GAMMA SAMPLING is not only significantly more efficient than K2T, which is also a guided decoding method, but also avoids the negative impact of over-control strength to a large extent.

5) In the main results, GAMMA SAMPLING-steered GPT2 generally outperforms all the representative baselines for controllable generation in terms of diversity, attribute relevance, and overall quality of generated samples.

6) In the detailed results (see Appendix D) of the human evaluation, GSM-XL (GAMMA SAMPLING-steered GPT2-XL) was the best in terms of controllable topic and controllable sentiment.

[1] A Plug-and-Play Method for Controlled Text Generation, 2021, Pascual et al.

---

### Decision · Program_Chairs · 2023-01-20

**Decision:**

Reject

**Justification For Why Not Higher Score:**

The authors present one way to do constrained generation (that is quite similar to prior work) and then evaluates their approach against completely different methods (omitting evaluation against similar work).

**Justification For Why Not Lower Score:**

N/A

**Metareview: Summary, Strengths And Weaknesses:**

The paper proposed Gamma Sampling (GS), a sampling method that enables controllable text generation for specific attributes defined by word lists. The key idea of this approach is to modify the logits/probability of the attribute-specific words by a temperature-like hyperparameter (called Gamma) and rescale the softmax distribution at each step. The main advantage of the GS is that it doesn’t require any condition-specific training data. The authors conduct experiments on six controllable generation tasks, and GS appears to be effective and efficient.

The authors revised the paper to address some of the original concerns of the reviewers, including small GPT2 model size, lack of comparisons against inference-time controllable text generation baselines, and lack of details on the human evaluation. This eased some of the reviewers’ concerns, and the reviewers increased some of their scores accordingly.

Ultimately the paper is still below borderline, as the reviewers still have the following concerns:

1. Impact:  The applicability of this work is relatively limited compared to prior controllable text generation methods, as GS is governed by word lists that need to be independently provided. So, GS cannot be used for tasks or goals that cannot easily be verbalized.
2. Novelty: The authors clarified the difference between GS and some of the related work mentioned in the reviews, but the reviewers still felt the level of novelty is relatively limited. The approach is to just heuristically adjust the probabilities of salient words/tokens, in a step-wise manner, an idea that has appeared in a number of prior work in some forms (details in the reviews).
3. Evaluation: While GS seems effective, the evaluation mostly compares GS to conditional/fine-tuned LM methods and comparisons against closely related and methodologically similar work are missing. Three reviewers had various comments that relate to this inadequacy of the evaluation, but the revised paper only added one model in response to these concerns (K2T, which wasn’t the one suggested by the reviewers). Given point (2) above, I think the paper should be more focused on evaluating the specific differences with similar approaches in prior work. If these approaches are not directly comparable in the evaluation setting of the paper, the authors could at least provide some relevant baselines (e.g., the naïve GS-related baseline suggested by Reviewer aPW9) or ablations of GS. In its current form, the paper evaluates GS mostly against methods that are quite different (and often not strictly comparable due to differences in model sizes) and the overall lack of analyzes makes it difficult to appreciate why GS works and what affects its effectiveness (e.g., one reviewer suggested evaluating the impact of number of control elements).

I recommend rejection in light of these concerns.

**Summary Of Ac-Reviewer Meeting:**

There was no meeting as this wasn't on the list of papers shared with the SAC. However, the average score bumped up to 5.25 a few days before the meta-reviews were due and there was no time to organize a meeting. However, I did communicate with the reviewers to converge on a recommendation. The only reviewer who gave a recommendation above acceptance threshold confirmed by email they are fine rejecting the paper, given the various remaining concerns discussed in the reviewer discussion in OpenReview.